# Hahahahaha, Duuuuude, Yeeessss!: A two-parameter characterization of stretchable words and the dynamics of mistypings and misspellings

Tyler J. Gray[1,2,3,4]*, Christopher M. Danforth[1,2,3,4]*, Peter Sheridan Dodds[1,2,3,4]*

**1** Department of Mathematics and Statistics, University of Vermont, Burlington, VT, United States of America, **2** Vermont Complex Systems Center, University of Vermont, Burlington, VT, United States of America, **3** Computational Story Lab, University of Vermont, Burlington, VT, United States of America, **4** Vermont Advanced Computing Core, University of Vermont, Burlington, VT, United States of America

* tyler.gray@uvm.edu (TJG); chris.danforth@uvm.edu (CMD); peter.dodds@uvm.edu (PSD)

**Data Availability Statement:** The data associated with this study can be downloaded from Twitter's public API (https://developer.twitter.com/en/docs.html). While Twitter's API terms of service will not

## Abstract

Stretched words like 'heellllp' or 'heyyyyy' are a regular feature of spoken language, often used to emphasize or exaggerate the underlying meaning of the root word. While stretched words are rarely found in formal written language and dictionaries, they are prevalent within social media. In this paper, we examine the frequency distributions of 'stretchable words' found in roughly 100 billion tweets authored over an 8 year period. We introduce two central parameters, 'balance' and 'stretch', that capture their main characteristics, and explore their dynamics by creating visual tools we call 'balance plots' and 'spelling trees'. We discuss how the tools and methods we develop here could be used to study the statistical patterns of mistypings and misspellings and be used as a basis for other linguistic research involving stretchable words, along with the potential applications in augmenting dictionaries, improving language processing, and in any area where sequence construction matters, such as genetics.

## 1 Introduction

Watch a soccer match, and you are likely to hear an announcer shout 'GOOOOOOOOOAAAAAAAAL!!!!!!'. Vowel lengthening and consonant lengthening (called gemination) is a feature of some languages and can change a word, including its meaning [1]. Stretched words, as in the example above, sometimes called elongated words [2], are also an integral part of many languages, especially in spoken language. However, rather than completely changing the meaning of the word, this stretching, also called word lengthening [3], expressive lengthening [4, 5], or use of letter repetitions [6], is often used to modify the meaning of the base word in some way, such as to strengthen the meaning (e.g., 'huuuuuge'), imply sarcasm (e.g., 'suuuuure'), show excitement (e.g., 'yeeeessss'), or communicate danger (e.g., 'nooooooooooooo'). We will refer to words that are amenable to such lengthening as 'stretchable words'.

allow us to share the individual messages we used, we used Twitter's gardenhose stream from 9 September 2008 to 31 December 2016, providing a 10% random sample of tweets for that time period, and any good sample of tweets over the same time period should provide similar results.

**Funding:** CMD and PSD were supported by National Science Foundation Grant Number IIS-1447634, and TJG, CMD, and PSD were supported by a gift from MassMutual. The funders had no role in study design, data collection and analysis, decision to publish, or preparation of the manuscript.

**Competing interests:** We have the following interests: TJG, CMD, and PSD were supported by a gift from MassMutual. There are no patents, products in development, or marketed products to declare. This does not alter our adherence to all of the PLOS ONE policies on sharing data and materials.

However, despite their being a fundamental part of spoken language, stretched words are rarely found in literature and lexicons: There is no 'hahahahahahaha' in the Oxford English Dictionary [7]. Book appearances are few, and only really occur in fictional dialogue [8]. However, with the advent and rise of social media, stretched words have finally found their way into large-scale written text.

With the increased use of social media comes rich datasets of a linguistic nature, granting science an unprecedented opportunity to study the everyday linguistic patterns of society. As such, in recent years there have been a number of papers published that have used data from social media platforms, such as Twitter, to study different aspects of language [3, 5, 9–23].

Some recent studies have also begun to look at stretchable words [3–6, 8, 18–20, 24–27]. In his paper about emoticons, Schnoebelen looked at the differences between Twitter users who include a nose with their emoticon faces and those who do not. He found that, in general, users with noseless emoticons tended to have less formal writing, including an increased use of stretched words compared to users who included noses with their emoticons. The users who included a nose tended to use more standard writing, including fewer stretched words [5].

Eisenstein listed stretched words as one of many types of "bad language" found on social media that cause issues when trying to process text, and commented on some of the issues with the current proposed methods of normalization and domain adaptation to help with language processing [4].

Brody and Diakopoulos looked at stretched words in a small Twitter dataset, finding they are quite abundant and that there is a strong correspondence between stretched words and words that provide sentiment. They also proposed a method of automatically finding and classifying new sentiment bearing words using this connection [3]. Some other studies have also looked at stretchable words in relation to sentiment analysis of text [18–20, 27].

Kalman and Gergle studied how stretched words serve as an analogue to nonverbal cues, such as phoneme extension, in an older set of email messages. They found that most of the stretched letters correspond to articulable phonemes and onomatopoeic words make up a sizable portion of their list of stretchable words [6]. However, preliminary results also show that these computer-mediated communication (CMC) cues are going through an evolutionary process wherein they are losing their direct link to nonverbal cues and are developing characteristics and an identity of their own [6, 8, 26].

In this paper, we begin a far more comprehensive study of stretchable words within social media. We use an extensive set of social media messages collected from Twitter—tweets—to investigate the characteristics of stretchable words used in this particular form of written language. We perform a thorough search for stretched words, allowing for many more possible ways of stretching words than the previous studies do, collecting a much larger and more complete set of stretchable words.

The tools and approach we introduce here allow us to discover some of the basic characteristics of stretchable words and form a foundation for further linguistic research. They also have many other potential applications, including the possible use by dictionaries to formally include this intrinsic part of language. The online dictionary Wiktionary has already discussed the inclusion of some stretched words and made a policy on what to include [28, 29]. Some other potential applications include the use by natural language processing software and toolkits, search engines, and by Twitter to build better spam filters.

We structure our paper as follows: In Sec. 2, we detail our dataset and our method of collecting stretchable words and distilling them down to their 'kernels'. In Sec. 3.1, we examine the frequency distributions for lengths of stretchable words. We quantify two independent properties of stretchable words: Their 'balance' in Sec. 3.2 and 'stretch' in Sec. 3.3. In Sec. 3.4, we develop an investigative tool, 'spelling trees', as a means of visualizing stretchable words

involving a two character repeated element. We comment on mistypings and misspellings in Sec. 3.5. Finally, in Sec. 4, we provide some additional discussion and concluding remarks.

## 2 Description of the dataset and method for extracting stretched words

The Twitter dataset we use in this study comprises a random sample of approximately 10% of all tweets (the 'gardenhose' API) from 9 September 2008 to 31 December 2016. In order to remain compliant with Twitter's API, we do not share the individual messages we use. However, any good sample of tweets over the same time period should provide similar results.

We limit our scope to tweets that either were flagged as an English tweet or not flagged for any language. All tweets in this time period have a maximum length of 140 characters. To collect stretchable words, we begin by making all text lowercase and collecting all tokens within our dataset from calendar year 2016 that match the Python regular expression r'(\b\w*(\w)(\w)(?:\2|\3){28,}\w*\b)'. This pattern will collect any token with at least 30 characters that has a single character repeated at least 29 times consecutively, or two different characters that are repeated in any order at least 28 times, for a total of at least 30 consecutive repeated occurrences of the two characters. The choice of 28 in the regular expression is a threshold we chose with the goal of limiting our collection to tokens of words that really do get stretched in practice.

After collecting these tokens, we remove any that contains a character that is not a letter ([a-z]), and distill each remaining token down to its 'kernel'. Table 1 gives a few examples of this distillation process. Proceeding along the token from left to right, whenever any pair of distinct letters, $l_1$ and $l_2$, occur in the token where (1.) $l_1$ occurs followed by any sequence of $l_1$ and $l_2$ of total length at least three, and (2.) such that $l_1$ and $l_2$ each occur at least twice in the sequence, we replace the sequence with the 'two letter element' ($l_1$ $l_2$). For example, see the first cell in Table 1.

In certain cases we distill the token to a kernel that is less general. These cases that are exceptions to the preceding are: (1.) The case where the sequence is a series of $l_1$ followed by a series of $l_2$, which is replaced with the pair of 'single letter elements' $[l_1][l_2]$. For example, see the second cell in Table 1. And (2.), the case where the sequence is a series of $l_1$ followed by a series of $l_2$ followed by a series of $l_1$, which is replaced with $[l_1][l_2][l_1]$. For example, see the first step in the fourth cell of Table 1 where 'bbbbbaaaaaabbbbbb' is replaced with [b][a][b].

**Table 1. Examples of distilling tokens down to their kernels.** The first line of each cell is the example token. The following lines show the result after every time a replacement of characters by the corresponding single letter element(s) or double letter element is made by the code, in order. The final line of each cell gives the resulting kernel for each example.

| | |
|---|---|
| 1. | hahhahahaahahaa<br>→ (ha) |
| 2. | gooooooaaaaaaal<br>→ g[o][a]l |
| 3. | ggggooooooaaaaalllll<br>→ [g][o]aaaaalllll<br>→ [g][o][a][l] |
| 4. | bbbbbaaaaaabbbbbbyyyyyyy<br>→ [b][a][b]yyyyyyy<br>→ [b][a][b][y] |
| 5. | awawawaaawwwwwessssssommmmmeeeeee<br>→ (aw)essssssommmmmeeeeee<br>→ (aw)esssssso[m][e]<br>→ (aw)e[s]o[m][e] |

Following this process, whenever a single letter, $l_3$, occurs two or more times in a row, we replace the sequence with the single letter element $[l_3]$. For example, see the last step of the fourth cell in Table 1 where 'yyyyyyy' is replaced with [y], or the last step in the fifth cell where 'sssss' is replaced with [s].

We collected tokens in batches of seven consecutive days at a time throughout 2016 (with the last batch being only two days). If a kernel is not found in more than one batch, or within the same batch but from at least two distinct stretched words, then it is removed from consideration.

Different but related stretched words (that is, different stretched words, but both stretched out versions of the same base word) may distill to different kernels. We combine these into a single, more general kernel for each word such that it covers all cases observed in the collected tokens. For example, for the two stretched versions of 'goal', 'goooalll' and 'goaaaalllllll', the first would distill to the kernel g[o]a[l] and the second would distill to go[a][l]. These two kernels would be combined as g[o][a][l].

Similarly, the kernels h[a] and (ha) would be combined as (ha) as the set of tokens represented by (ha) is a superset of the set of the tokens represented by h[a]. Tokens that match h[a] must have one 'h' followed by one or more 'a's whereas tokens that match (ha) are anything that start with an 'h' that is then followed by any number of 'h's and 'a's, in any order, as long as there is at least one 'a'.

After processing our dataset, we obtained a collection of 7,526 kernels. We then represented each kernel with a corresponding regular expression and collected all tokens in our entire gardenhose dataset that matched the regular expressions. To go from the kernel to the regular expression, we replaced] with]+, replaced $(l_1 l_2)$ with $l_1 [l_1 l_2]^* l_2 [l_1 l_2]^*$, and we surrounded the kernel with word boundary characters \b. So, for example, the kernel g[o][a][l] goes to the Python regular expression r'\bg[o]+[a]+[l]+\b' and the kernel (ha) goes to the Python regular expression r'\bh[ha]*a[ha]*\b'.

Once we collected all tokens matching our kernels, we carried out a final round of thresholding on our kernel list, removing those with the least amount of data and least likely to represent a bona fide stretchable word. For each kernel, we calculated the token count as a function of token length (number of letters) for all tokens matching that kernel. For example, Fig 1 gives the plot of the token count distribution for the kernel (to). Then, with the token counts in order by increasing token length, as in Fig 1, we found the location of the largest drop in the $\log_{10}$ of token counts between two consecutive values within the first 10 values. That is, if we let $f_l$ be the token count for tokens of length $l$, and let the kernel length (smallest token length) be $\ell$, then we find the token length, $l_{\mathrm{drop}}$, where this largest drop occurs as

$$l_{\mathrm{drop}} = \operatorname*{argmax}_{\ell \leq l \leq \ell+9} \ \log_{10} f_l - \log_{10} f_{l+1}. \tag{1}$$

We call the words with lengths coming before the location of the drop ($l \leq l_{\mathrm{drop}}$) 'unstretched' versions of the kernel and those that come after ($l > l_{\mathrm{drop}}$) 'stretched' versions. For most kernels, the largest drop will be between the first and second value. However, for some kernels this drop occurs later. For example, in Fig 1 we see that for the kernel (to), which covers both the common words 'to' and 'too', this drop is between the second and third value (between tokens of length three and four; $l_{\mathrm{drop}} = 3$). Thus, the unstretched versions of (to) are represented by the first two points in Fig 1, with the remaining points representing stretched versions of (to).

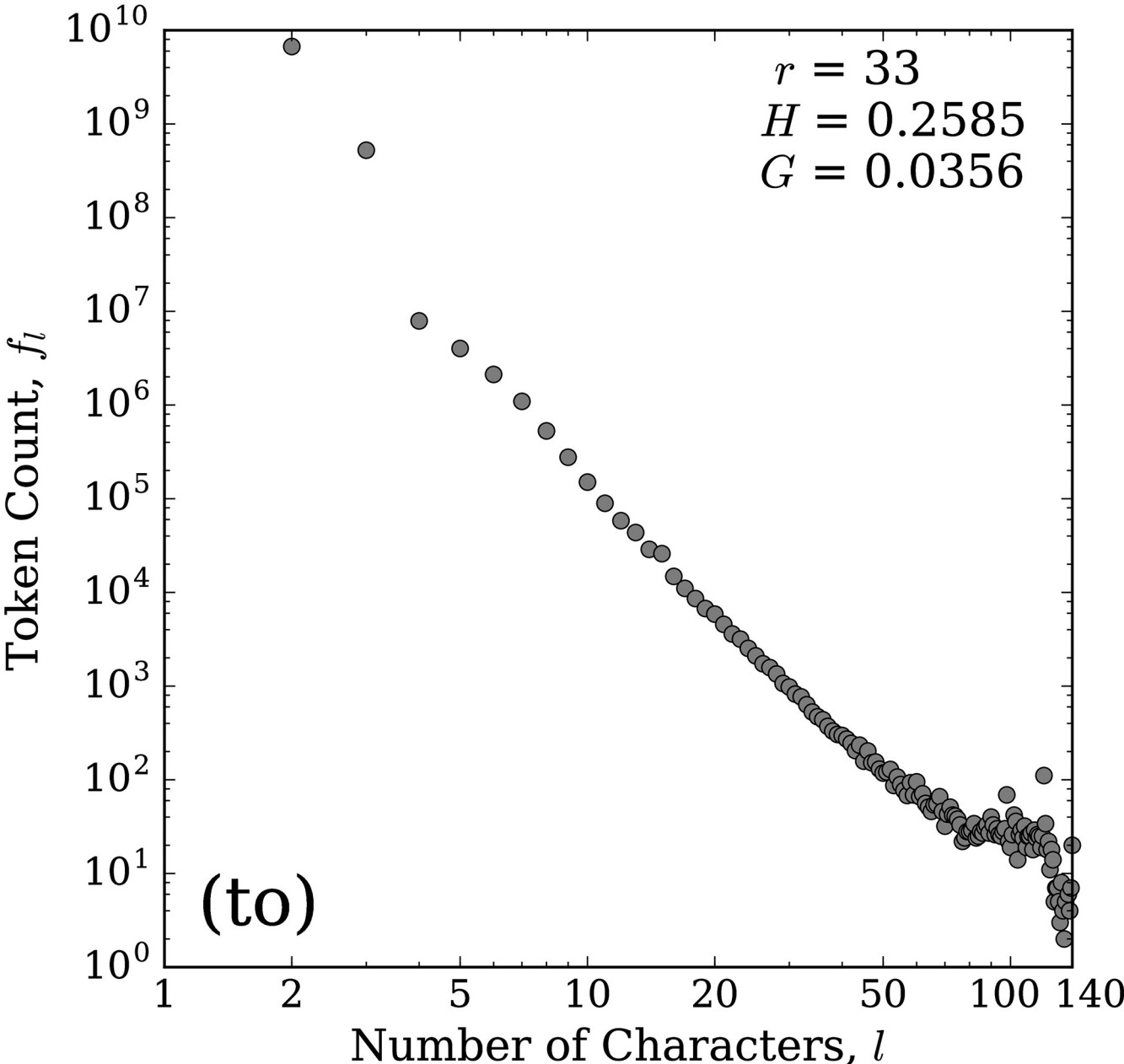

**Fig 1. Token count distribution for the kernel (to).** The horizontal axis represents the length (number of characters), $l$, of the token and the vertical axis gives the total number of tokens of a given length that match this kernel, $f_l$. The included statistics give the kernel rank, $r$ (see Sec. 2), the value of the balance parameter (normalized entropy, $H$; see Sec. 3.2), and the value of the stretch parameter (Gini coefficient, $G$; see Sec. 3.3) for this kernel. The large drop between the second and third points denotes the change from 'unstretched' versions of (to), located to the left of this drop, to 'stretched' versions of (to), located to the right of this drop.

We then ranked the kernels by the sum of the token counts for their stretched versions,

$$n_\text{s} = \sum_{l=l_\text{drop}+1}^{140} f_l. \tag{2}$$

Fig 2 shows this sum as a function of rank for each kernel. Inspired by the idea of a cutoff frequency [30], we estimate a cutoff rank for the kernels. Using the values between rank 10 and $10^3$, we found the regression line between the $\log_{10}$ of the ranks and the $\log_{10}$ of the summed token counts (straight line, Fig 2). We calculated the cutoff as the first rank (after $10^3$) where the summed token count is less than 1/10 of the corresponding value of the regression line. That is, if we let $\alpha$ and $\beta$ be the slope and intercept paramters calculated during regression, and $n_{s,r}$ be the total number of stretched tokens for the kernel at rank $r$, then we find the smallest $r$ such that $r > 10^3$ and

$$n_{s,r} < \frac{10^\beta r^\alpha}{10} \tag{3}$$

as our rank cutoff. This occurs at rank 5,164, which is shown by the vertical dashed line in Fig 2. For the remainder of this study, we used the kernels with rank preceding this cutoff, giving us a total of 5,163 kernels, and, unless otherwise specified, a kernel's 'rank', $r$, refers to the rank found here.

Note that we are using a rough guide to find a practical cutoff for the number of kernels we include in our study. While we are finding a linear fit as part of this process, this token count distribution is not some archetypal power-law. We merely use the regression line as a reference from which to calculate a drop analogous to the process of finding a cutoff frequency, and the precise cutoff is not particularly important. The cutoff rank is not used in the statistics of any individual kernel, and for the analyses that examine how stretchable words behave as a function of kernel rank, the resultant figures and statistics will only be affected at the margin of the cutoff rank. An alternative might be to simply pick a cutoff rank based on visual inspection of Fig 2 or to pick a lower bound for the data amount (token count sum, $n_s$), and find which rank falls below that bound.

See Online Appendix A at http://compstorylab.org/stretchablewords/ for a full list of kernels meeting our thresholds, along with their regular expressions and other statistics discussed throughout the remainder of this paper.

## 3 Analysis and results

### 3.1 Distributions

For each kernel, we plotted the corresponding distribution of token counts, $f_l$, as a function of token length, $l$. Most of the distributions largely follow a roughly power-law shape. For example, Fig 3 gives the frequency distribution for the kernel [g][o][a][l]. From the elevated frequency of the first dot, we can see that the unstretched word 'goal' is used about two orders of magnitude more frequently than any stretched version. After the first point, we see a rollover in the distribution, showing that if users are going to stretch the word, they are more likely to include a few extra characters rather than just one. We also see that there are some users who indeed fill the 140 character limit with a stretched version of the word 'goal', and the elevated dot there suggests that if users get close to the character limit, they are more likely to fill the available space. The other dots elevated above the trend represent tokens that likely appear in tweets that have a small amount of other text at the beginning or end, such as a player name or team name, or, more generally, a link or a user handle.

In Fig 4, we show the frequency distribution for the kernel (ha) as an example of a distribution for a two character repeated element. For this distribution we observe an alternating up and down in frequency for even length tokens and odd length tokens. This behaviour is typical of distributions with a two character repeated element, likely resulting from an intent for these tokens to be a perfect alternating repetition of 'h' and 'a', hahaha. . ., to represent laughter.

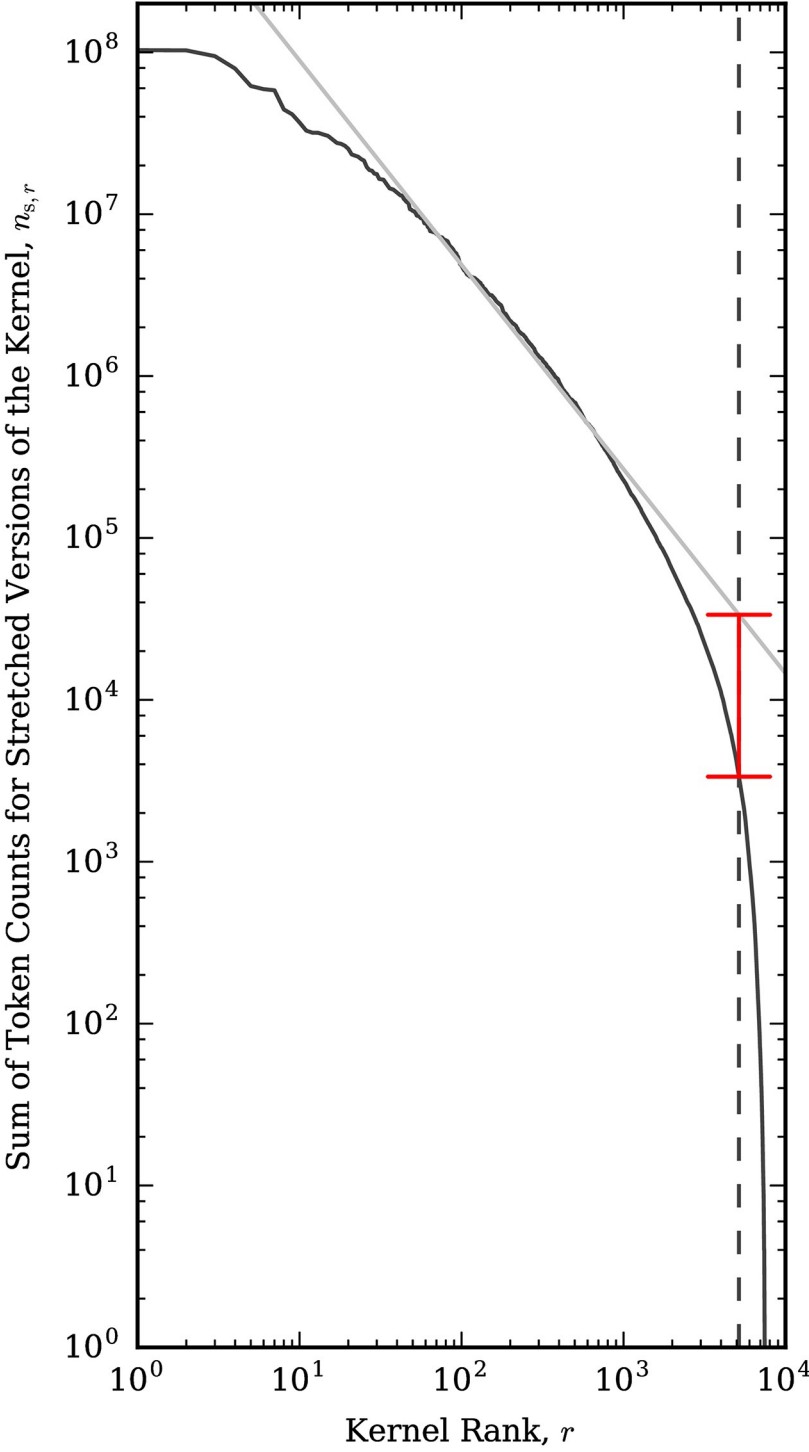

**Fig 2. Total token counts for stretched versions of all kernels.** Kernels are ranked by their descending total token count along the horizontal axis. The diagonal line gives the regression line calculated using the values between ranks 10 and $10^3$. The vertical dashed line denotes the first location after rank $10^3$ where the distribution drops below 1/10 of the corresponding value of the regression line, denoted by the red interval, giving the cutoff rank for the final threshold to decide which kernels to include in this study.

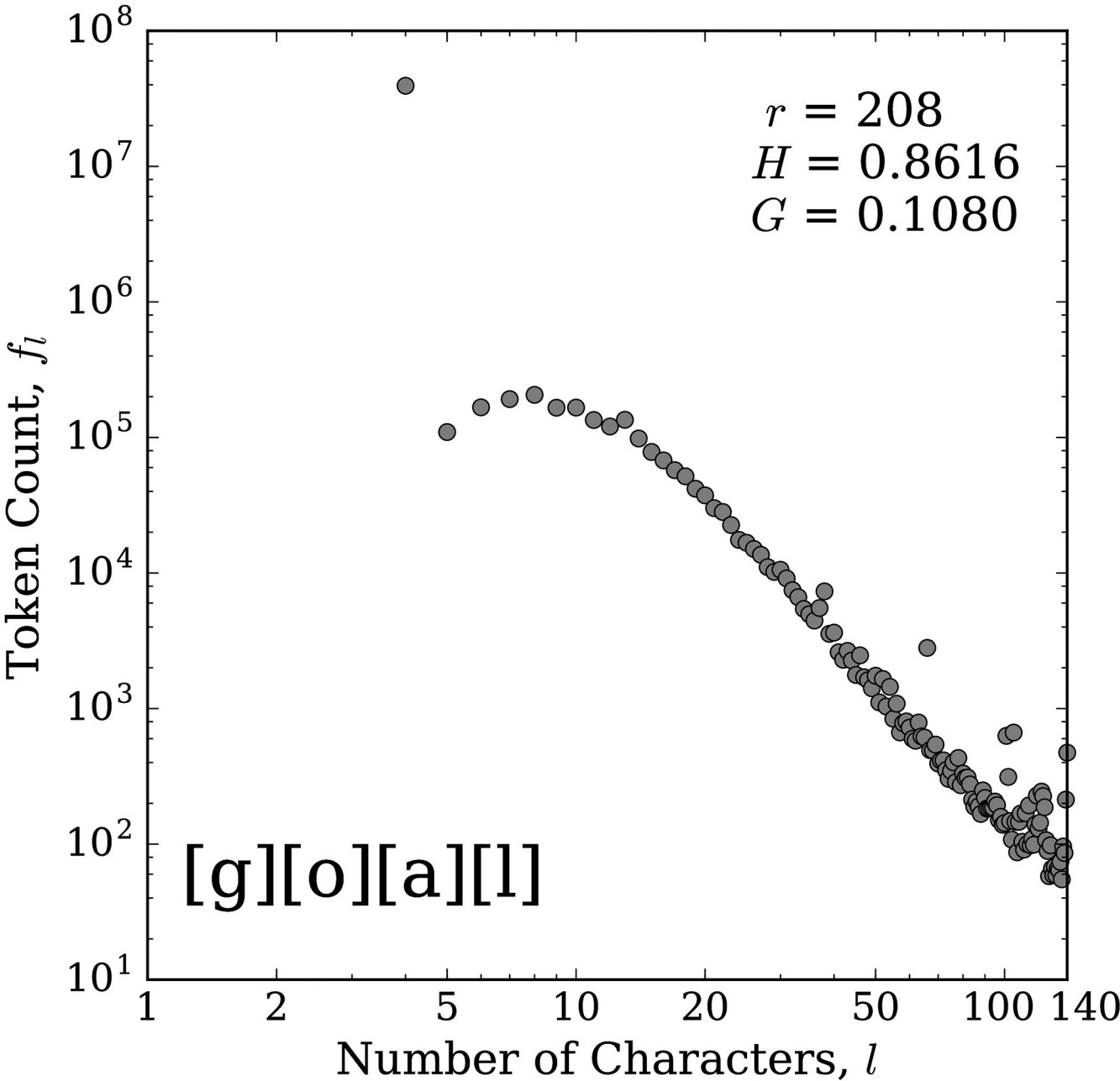

**Fig 3. Token count distribution for the kernel [g][o][a][l].** The horizontal axis represents the length (number of characters), $l$, of the token and the vertical axis gives the total number of tokens of a given length that match this kernel, $f_l$. See Fig 1 caption for details on the included statistics. The base version of the word appears roughly 100 times more frequently than the most common stretched version.

Under this assumption, the correct versions will be even length. Then, any incorrect version could be odd or even length depending on the number of mistakes. We look at mistakes further in Sec. 3.5.

We note that there is also an initial rollover in this distribution, showing that the four character token, with dominant contributor 'haha', is the most common version for this kernel. We also again see some elevated counts near the tail, including for 140 characters, along with

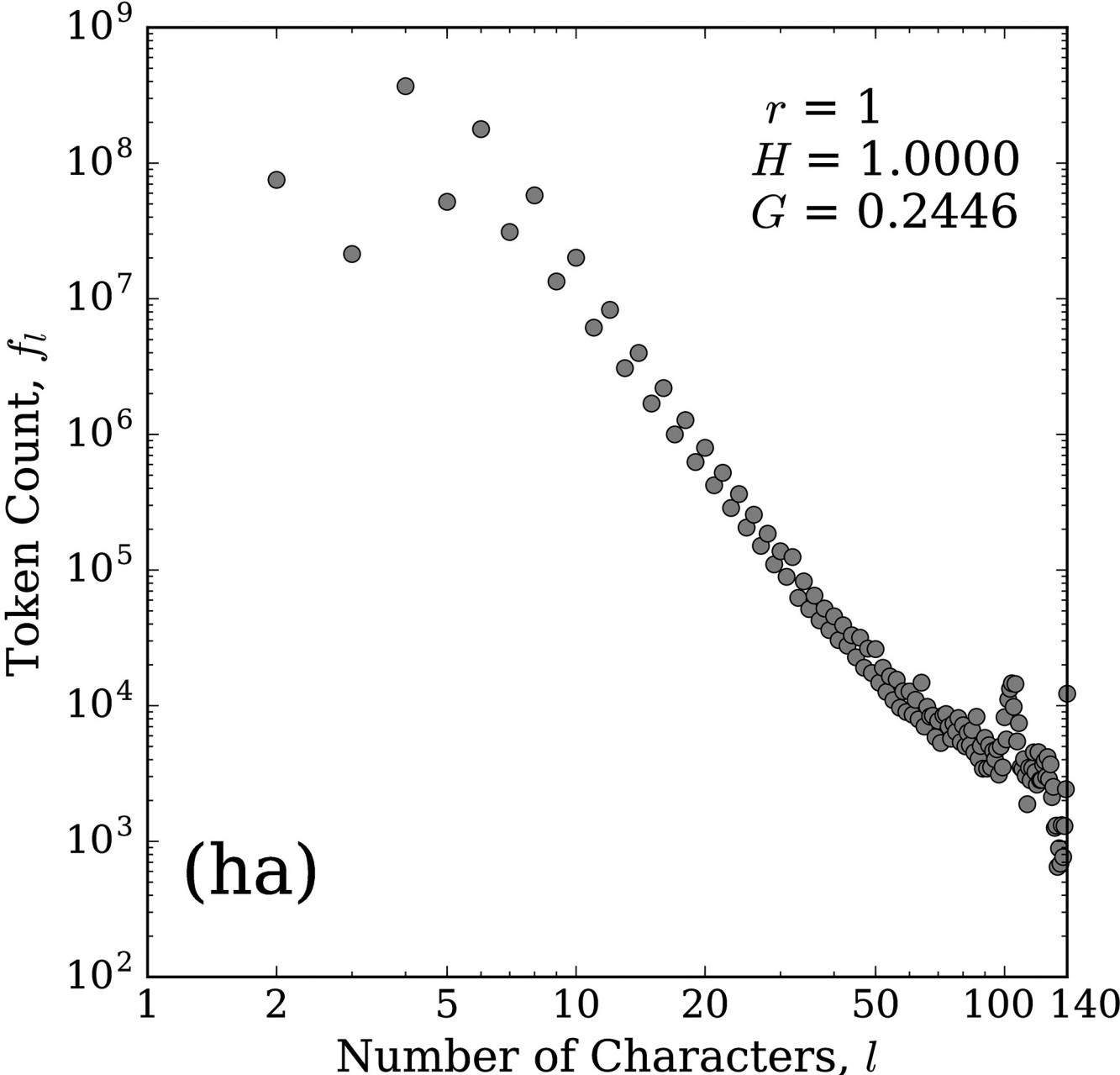

**Fig 4. Token count distribution for the kernel (ha).** The horizontal axis represents the length (number of characters), *l*, of the token and the vertical axis gives the total number of tokens of a given length that match this kernel, $f_l$. See Fig 1 caption for details on the included statistics.

some depressed counts just short of 140, which again suggests that when users approach the character limit with stretched versions of (ha), they will most likely fill the remaining space. We did not perform a detailed analysis of this area, but it is likely that the other elevated points near the end are again due to the inclusion of a link or user handle, etc. Similarly, the general flattening of the distribution's right tail is likely a result of random lengths of short other text combined with a stretched word that fills the remaining space.

These distributions tend to generally follow Zipf's brevity law, or law of abbreviation, which states that more frequent words tend to be shorter [31, 32]. This law has been found to hold for many different languages, and possibly even for communication between other primates [31–36]. However, we find this law is not always strictly followed. Many of the distributions do have a rollover for the shorter word lengths, as seen in the two examples shown in Figs 3 and 4. If the brevity law is a result due in part to efficiency, then our counterexample observations may simply imply the existence of another constraint being optimized. Perhaps the rollover results from a balance between efficiency (keeping the word short) and novelty (stretching to distinguish from the base word). Additional letters avoid the appearance of a mistyping and make the word stand out more visually.

Similar distributions for each kernel can be found in Online Appendix B at http://compstorylab.org/stretchablewords/.

## 3.2 Balance

For each kernel, we measure two quantities: (1.) The balance of the stretchiness across characters, and (2.) the overall stretchiness of the kernel. To measure balance, we calculate the average stretch of each character in the kernel across all the tokens within a bin of token lengths. By average stretch of a character, we mean the average number of times that character appears. That is, if we let $c_{i,j,k}$ be the number of times character $i$ was repeated in token $k$ of bin $j$ and let $N_j$ be the number of tokens in bin $j$, then the average stretch of character $i$ in bin $j$, $\bar{c}_{i,j}$, is given by

$$\bar{c}_{i,j} = \frac{\sum_{k=1}^{N_j} c_{i,j,k}}{N_J}.$$ (4)

Fig 5 shows the balance for the kernel [g][o][a][l] partitioned into bins of logarithmically increasing sizes of length. The horizontal dashed lines represent the bin edges. The distance between the solid diagonal lines represents the average stretch, or average number of times each character was repeated, $\bar{c}_{i,j}$, and are plotted in the same order that they appear in the kernel. From this figure we see that 'g' is not stretched much on average, 'o' is stretched the most, and 'a' and 'l' are both stretched around 2/3 as much as 'o'.

When part of the kernel is a two letter element of the form $(l_1 l_2)$, we still count the number of occurrences of $l_1$ and $l_2$ corresponding to this element in the kernel separately, even though the letters can be intermingled in the stretched word. When we display the results, we display it in the same order that the letters appear in the kernel. So in Fig 6, which shows the results for the kernel (ha), the first space represents the average stretch for 'h' and the second space is for 'a'. From this figure, we can see that the stretch is almost perfectly balanced between the two letters on average.

Similar balance plots can be found for each kernel in Online Appendix C at http://compstorylab.org/stretchablewords/. In general, for these balance plots, we stop plotting at the first bin with no tokens, even if later bins may be nonempty.

For each kernel, we also calculate an overall measure of balance. To do this, we begin by binning the tokens by length, where unlike for the balance plots, each length is its own bin; we do not group multiple lengths into the same bin here. Then, for each bin (containing tokens longer than the kernel) we calculate the average stretch for each character across tokens within the bin, $\bar{c}_{i,j}$ as before. Then, we subtract one from each of these values (removing the contribution from each base character; counting just the number of times each character was repeated)

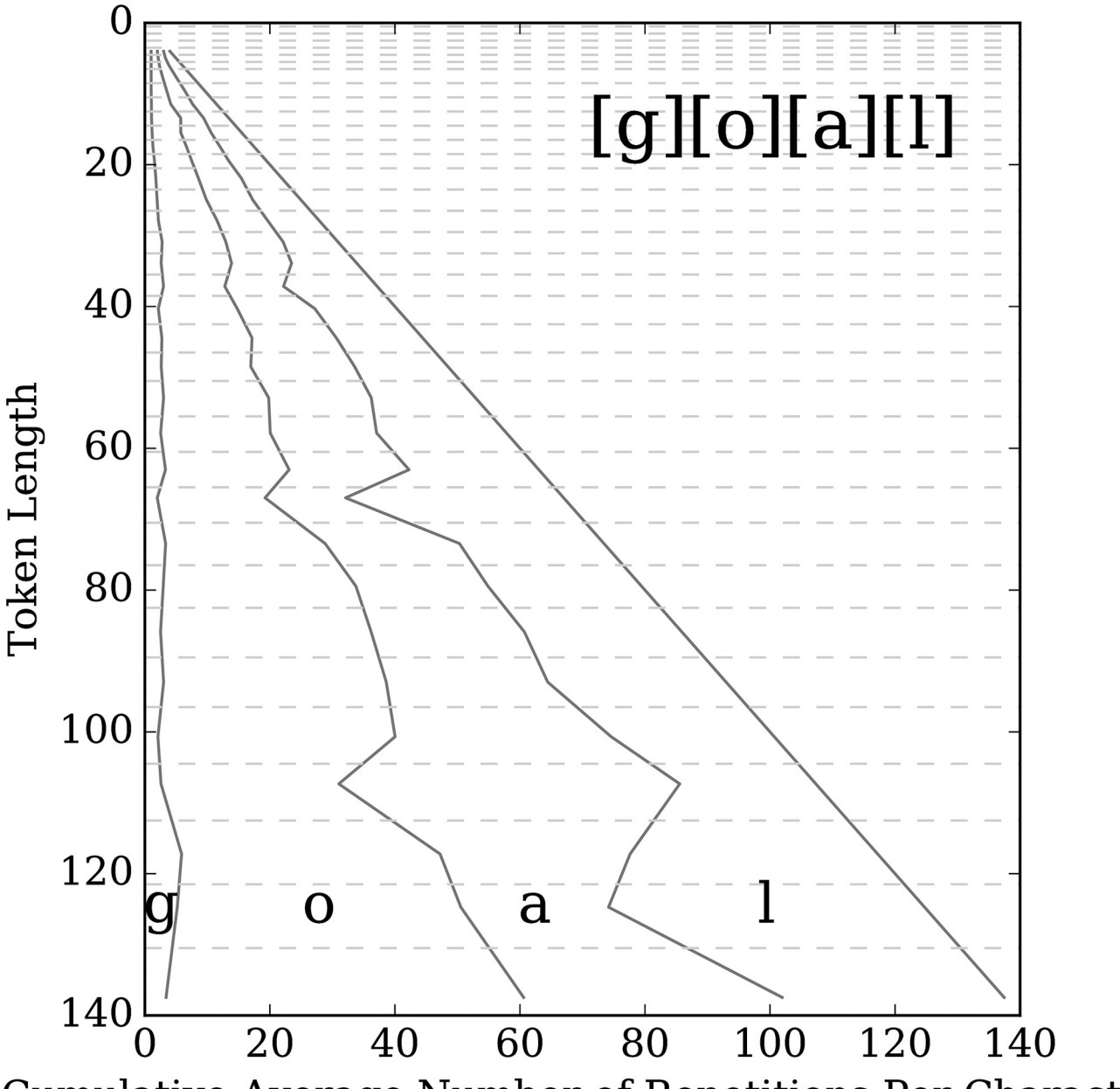

**Fig 5. Balance plot for the kernel [g][o][a][l].** The vertical axis represents the length (number of characters) of tokens, and is broken into bins of lengths, with boundaries denoted by horizontal dashed lines, which increase in size logarithmically. For all the tokens that match the kernel and fall within a bin of lengths, the average number of times each character was stretched in those tokens was calculated, and is shown on the plot as the distance between two solid lines in the same order as in the kernel. Thus, for a given bin, the distance between the vertical axis and the first solid line is the average stretch for the letter 'g', the distance between that first line and the second line is the average stretch for the letter 'o', and so on. For example, the last bin contains tokens with lengths in the interval [131, 140], with average length roughly 137. On average, tokens falling in this most celebratory bin contain roughly 3 'g's, 57 'o's, 41 'a's, and 36 'l's.

and normalize the values so they sum to 1 and can be thought of like probabilities,

$$p_{i,j} = \frac{\bar{c}_{i,j} - 1}{\sum_{i=1}^{\ell} (\bar{c}_{i,j} - 1)}, \tag{5}$$

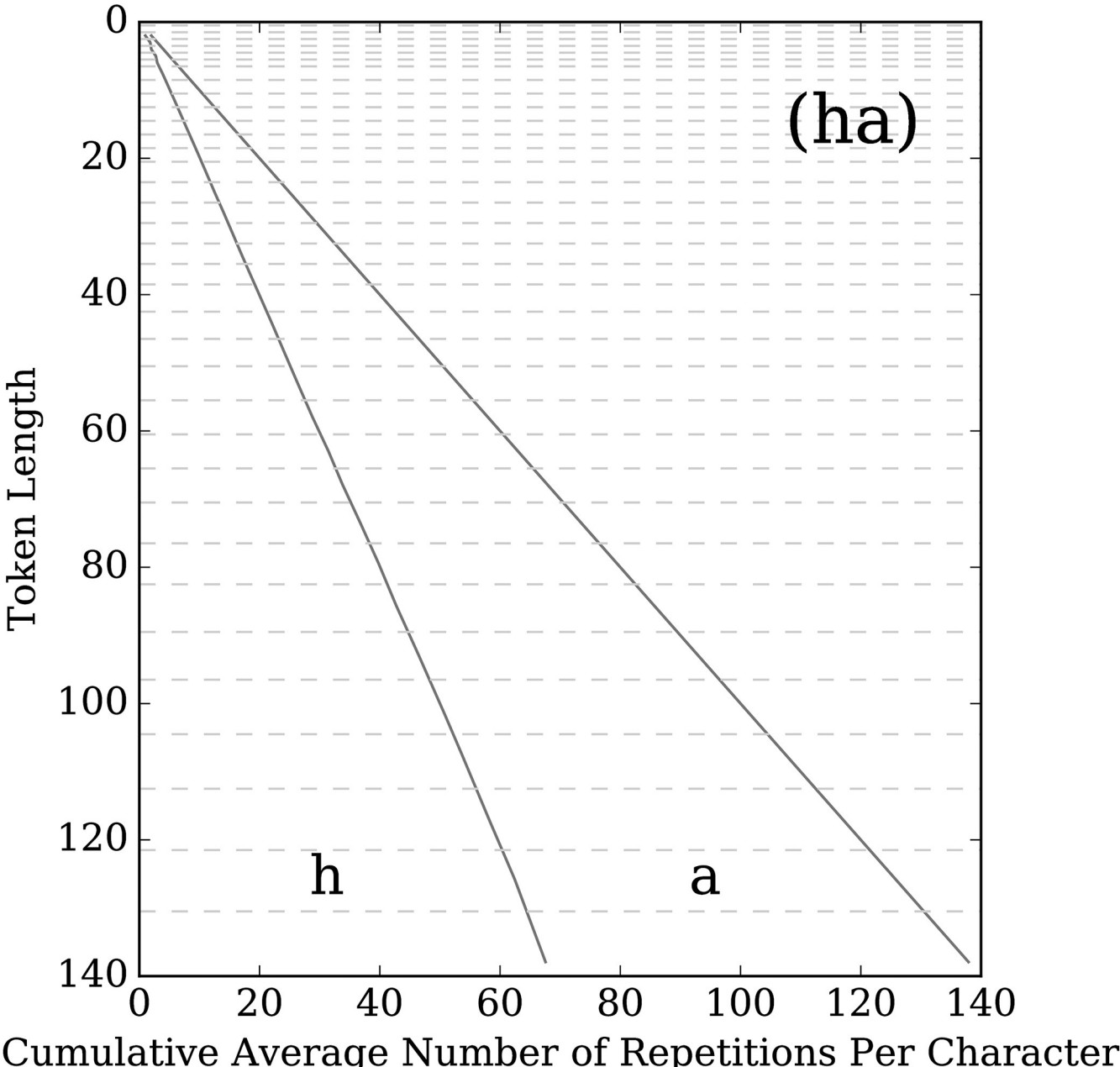

**Fig 6. Balance plot for the kernel (ha).** See the Fig 5 caption for plot details. For two letter elements, even though the letters can alternate within a given token, we still count the number of occurrences for each letter separately and display the average number of total repetitions in the same order as the letters appear in the kernel. Thus, for a given bin, the distance between the vertical axis and the first line is the average number of times the letter 'h' occurred in the tokens, and the distance between that first line and the second line is the average number of times the letter 'a' occurred in the token. This plot clearly shows that (ha) is well balanced across all bins of token lengths.

where $\ell$ is the number of characters in the kernel. We then average the probabilities across the bins, weighing each bin equally,

$$\bar{p}_i = \frac{\sum_{j=1}^{b} p_{i,j}}{b},$$

(6)

where $b$ is the number of bins. Finally, as our overall measure of balance, we compute the normalized entropy, $H$, of the averaged probabilities,

$$H = \frac{-\sum_{i=1}^{\ell} \bar{p}_i \log_2 \bar{p}_i}{\log_2 \ell}. \tag{7}$$

This measure is such that if each character stretches the same on average, the normalized entropy is 1, and if only one character in the kernel stretches, the normalized entropy is 0. Thus, higher entropy corresponds with more balanced words. (For a comparison with an alternate entropy measure where tokens contribute equally rather than equally weighing each length bin, and an explanation of the different corresponding views, see S1 Appendix).

Fig 7 shows two 'jellyfish plots' [37] for balance. Fig 7A is the version containing all words and for Fig 7B we remove the words that have a value of exactly 0 for entropy. The top of the plot in Fig 7A shows the frequency histogram of the normalized entropy for each kernel. The spike containing value 0 comes largely from kernels where only one character stretches, giving that kernel an entropy of exactly 0. The main plot shows the normalized entropy values as a function of word rank, where rank is given, as before, by the sum of stretched token counts. The 'tentacles' give rolling deciles. That is, for rolling bands of 500 words by rank, the deciles 0.1, 0.2, . . ., 0.9 are calculated for the entropy values, and are represented by the solid lines.

These jellyfish plots are useful in that they not only show the full frequency distribution, as provided by the histogram on their tops, but also allow us to see the stability of that distribution across ranks. The tentacle part of the plots allows us to see if highly ranked, more common kernels are distributed similarly to low ranked, less common kernels (tentacles fall fairly straight down), or if the kernels have different characteristics at different ranks (tentacles tend to drift left or right).

We can see from Fig 7A that the distribution largely shifts towards smaller entropy values with increasing rank, mostly drawn in that direction by the kernels with only a single letter that repeats and thus entropy exactly 0. For Fig 7B, we remove all kernels with entropy 0. Everything else remains the same, including the rank of each kernel (we skip over ranks of removed kernels) and the rolling bands of 500 kernels for percentile calculations still have 500 kernels, and thus tend to be visually wider bands. In contrast to Fig 7A, we now see a small left-shift in the earlier ranks, and then the distribution tends to stabilize for lower ranks. This shows that the highest ranked kernels tend to have a larger entropy, meaning the stretch of the kernel is more equally balanced across all characters. We also see that not many of the high ranked words stretch with just one character. It appears that these kernels that stretch in only a single character become more prevalent in the lower ranks.

Table 2 shows the kernels with the ten largest entropies and Table 3 shows those with the ten smallest nonzero entropies. We observe that the kernels with largest entropies are mostly of the form $(l_1 \, l_2)$ and are almost perfectly balanced. The least balanced kernels tend to be more recognizable English or Spanish words and names, with one exclamation also appearing in the bottom ten.

### 3.3 Stretch

To measure overall stretchiness for a kernel we calculated the Gini coefficient, $G$, of the kernel's token length frequency distribution. (For a comparison with another possible measure of stretch, see S2 Appendix) If the distribution has most of its weight on the short versions and not much on stretched out versions, then the Gini coefficient will be closer to 0. If more tokens are long and the kernel is stretched longer more often, the Gini coefficient will be closer to 1. Fig 8 gives the jellyfish plot for the Gini coefficient for each kernel. The horizontal axis has a

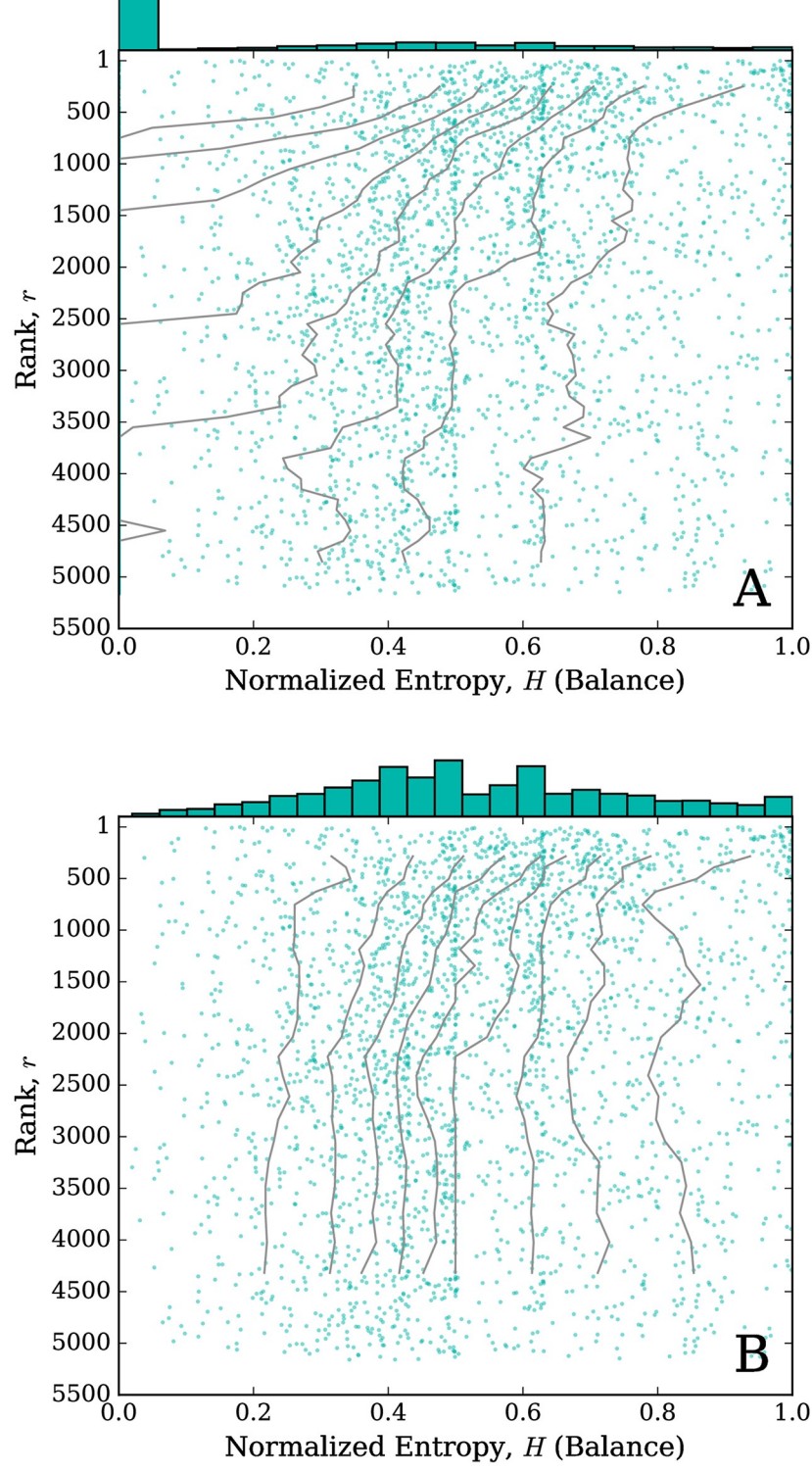

**Fig 7. Jellyfish plots for kernel balance for (A) all kernels, and (B) excluding kernels with entropy exactly 0.**
Corresponding histograms are given at the top of each plot. Kernels are plotted vertically by their rank, $r$, and horizontally by their balance as given by normalized entropy, $H$, where larger entropy denotes increased balance. The deciles 0.1, 0.2, . . ., 0.9 are calculated for rolling bins of 500 kernels and are plotted as the 'tentacles'.

**Table 2. Top 10 kernels by normalized entropy, *H*.**

|  | *H* | Kernel | Example token |
|---|---|---|---|
| 1 | 0.99998 | (kd) | kdkdkdkdkdkd |
| 2 | 0.99998 | (ha) | hahahahahaha |
| 3 | 0.99997 | [i][d] | iiiiiiiddddd |
| 4 | 0.99997 | (ui) | uiuiuiuiuiui |
| 5 | 0.99997 | (ml) | mlmlmlmlmlmlml |
| 6 | 0.99995 | (js) | jsjsjsjsjsj |
| 7 | 0.99990 | [e][t] | eeeeettttt |
| 8 | 0.99988 | (ox) | oxoxoxoxoxox |
| 9 | 0.99980 | (xq) | xqxqxqxqxqxqxq |
| 10 | 0.99971 | (xa) | xaxaxaxaxaxa |

logarithmic scale, and the histogram bins have logarithmic widths. From this plot, we see that the distribution for stretch is quite stable across word ranks, except for perhaps a slight shift towards higher Gini coefficient (more stretchiness) for the highest ranked kernels.

Table 4 shows the top 10 kernels ranked by Gini coefficient and Table 5 shows the bottom 10. The top kernel is [k], which represents laughter in Portuguese, similar to (ha) in English (and other languages). Containing a single letter, [k] is easier to repeat many times, and does not have an unstretched version that is a common word. We also see (go)[l] on the list, where 'gol' is Spanish and Portuguese for 'goal'. Interestingly, (go)[l] has a much higher Gini coefficient ($G = 0.5171$) than [g][o][a][l] does ($G = 0.1080$). The kernels with lowest Gini coefficient all represent regular words and all allow just one letter to stretch, which does not get stretched much.

In Fig 9, we show a scatter plot of each kernel where the horizontal axis is given by the measure of balance of the kernel using normalized entropy, and the vertical coordinate is given by the measure of stretch for the kernel using the Gini coefficient. Thus, this plot positions each kernel in the two dimensional space of balance and stretch. We see that the kernels spread out across this space and that these two dimensions capture two independent characteristics of each kernel.

We do note that there are some structures visible in Fig 9. There is some roughly vertical banding. In particular, the vertical band at $H = 0$ is from kernels that only allow one character to stretch and the vertical band near $H = 1$ is from kernels where all characters are allowed to stretch and do so roughly equally, which especially occurs with kernels that are a single two

**Table 3. Bottom 10 kernels by normalized entropy, *H*.**

|  | *H* | Kernel | Example token |
|---|---|---|---|
| 1 | 0.01990 | [b][o][b]ies | boooooobies |
| 2 | 0.02526 | [d][o][d]e | dooooooode |
| 3 | 0.03143 | infini[t][y] | infinityyyy |
| 4 | 0.03342 | che[l]se[a] | chelseaaaaaa |
| 5 | 0.03587 | tay[l]o[r] | taylorrrrrr |
| 6 | 0.03803 | f(re) | freeeeeeeeeeee |
| 7 | 0.03930 | [f]ai[r] | fairrrrrrr |
| 8 | 0.05270 | regr[e][s][e] | regreseeeee |
| 9 | 0.05271 | herm[a][n][a] | hermanaaaaaaa |
| 10 | 0.05323 | sq[u][e] | squueeeeeee |

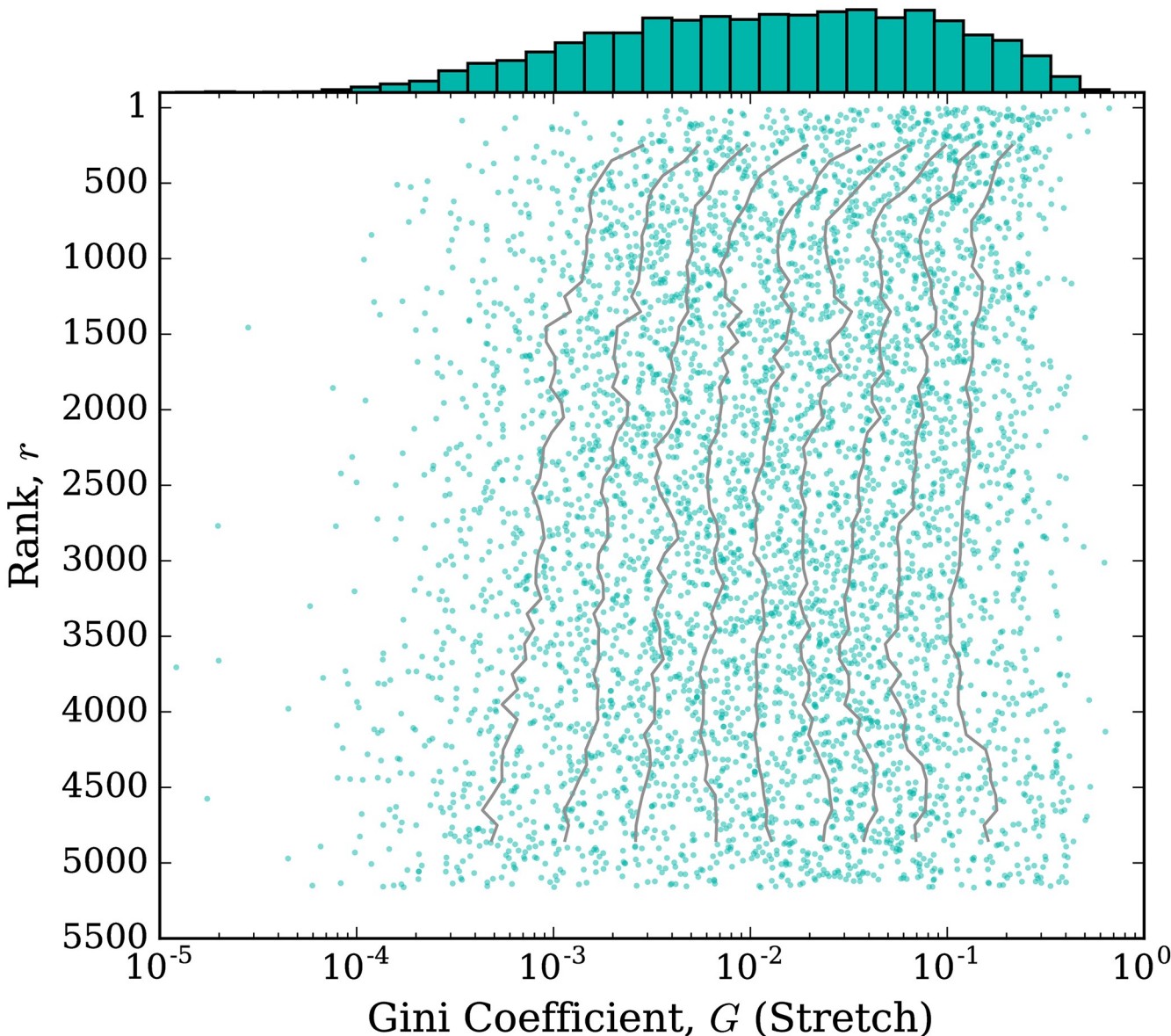

**Fig 8. Jellyfish plots for kernel stretch as measured by the Gini coefficient, *G*, of its token count distribution, where higher Gini coefficient denotes increased stretch.** The histogram is given at the top of the plot (with logarithmic width bins). Kernels are plotted vertically by their rank, *r*, and horizontally (on a logarithmic scale) by their stretch. The deciles 0.1, 0.2, . . ., 0.9 are calculated for rolling bins of 500 kernels and are plotted as the 'tentacles'.

letter element. Fainter banding around $H \approx .43$, $H \approx .5$, and $H \approx .63$ can also be seen. This largely comes from kernels of length 5, 4, and 3, respectively, that allow exactly two characters to stretch and those characters stretch roughly equally. If the stretch was perfectly equal, then the normalized entropy in each respective case would be $H = 1/\log_2(5) \approx .43$, $H = 1/\log_2(4) = .5$, and $H = 1/\log_2(3) \approx .63$.

### 3.4 Spelling trees

So far we have considered frequency distributions for kernels by token length, combining the token counts for all the different words of the same length matching the kernel. However,

**Table 4. Top 10 kernels by Gini coefficient, *G*.**

|  | *G* | Kernel | Example token |
|---|---|---|---|
| 1 | 0.66472 | [k] | kkkkkkkkkkkkkkk |
| 2 | 0.63580 | [w][v][w] | wwwwwwwwwwwvwwww |
| 3 | 0.62843 | [m][n][m] | mmmmmmmmmmmmnm |
| 4 | 0.53241 | [o][c][o] | ooooooooooco |
| 5 | 0.52577 | wa(ki) | wakikikikkkikikik |
| 6 | 0.51706 | (go)[l] | goooooooooool |
| 7 | 0.51273 | [m][w][m] | mmmmmwmmmmmmmmm |
| 8 | 0.50301 | galop[e]ir[a] | galopeeeeira |
| 9 | 0.50193 | [k][j][k] | kkkkkjjkkkkkkkkkk |
| 10 | 0.49318 | [i][e][i] | iiiiiieeiiiiiii |

different tokens of the same length may of course be different words—different stretched versions—of the same kernel. For kernels that contain only single letter elements, these different versions may just have different amounts of the respective stretched letters, but all the letters are in the same order. However, for kernels that have two letter elements, the letters can change order in myriad ways, and the possible number of different stretched versions of the same length becomes much larger and potentially more interesting.

In order to further investigate these intricacies, we introduce 'spelling trees' to give us a visual method of studying the ways in which kernels with two letter elements are generally expanded. Fig 10 gives the spelling tree for the kernel (ha). The root node is the first letter of the two letter element, which in this case is 'h'. Then, recursively, if the next letter in the word matches the first letter of the pair, it branches left, represented by a lighter gray edge, and if it matches the second letter of the pair then it branches right, represented by a darker gray edge. This branching continues until the word is finished. The first few nodes are highlighted with the letter corresponding to that point of the tree. The edge weights are logarithmically related to the number of tokens flowing through them. So, a thicker edge represents that more tokens pass through that edge than a thinner edge does. In Fig 10, a few nodes, denoted by stars, are annotated with the exact word to which they correspond. The annotated nodes are all leaf nodes, but words can, and most do, stop at nodes that are not leaves. We also trimmed the tree by only including words that have a token count of at least 10,000. This threshold of pruning reveals the general pattern while avoiding making the spelling tree cluttered.

**Table 5. Bottom 10 kernels by Gini coefficient, *G*.**

|  | *G* | Kernel | Example token |
|---|---|---|---|
| 1 | 0.00001 | am[p] | amppppppppp |
| 2 | 0.00002 | m[a]kes | maaaaaaaaakes |
| 3 | 0.00002 | fr[o]m | frooooooooooom |
| 4 | 0.00002 | watch[i]ng | watchiiiiiing |
| 5 | 0.00003 | w[i]th | wiiiiiiiith |
| 6 | 0.00004 | pla[y]ed | playyyyyyed |
| 7 | 0.00004 | s[i]nce | siiiiiiiince |
| 8 | 0.00006 | eve[r]y | everrrrrrrrry |
| 9 | 0.00006 | manage[r] | managerrrrr |
| 10 | 0.00007 | learnin[g] | learninggggg |

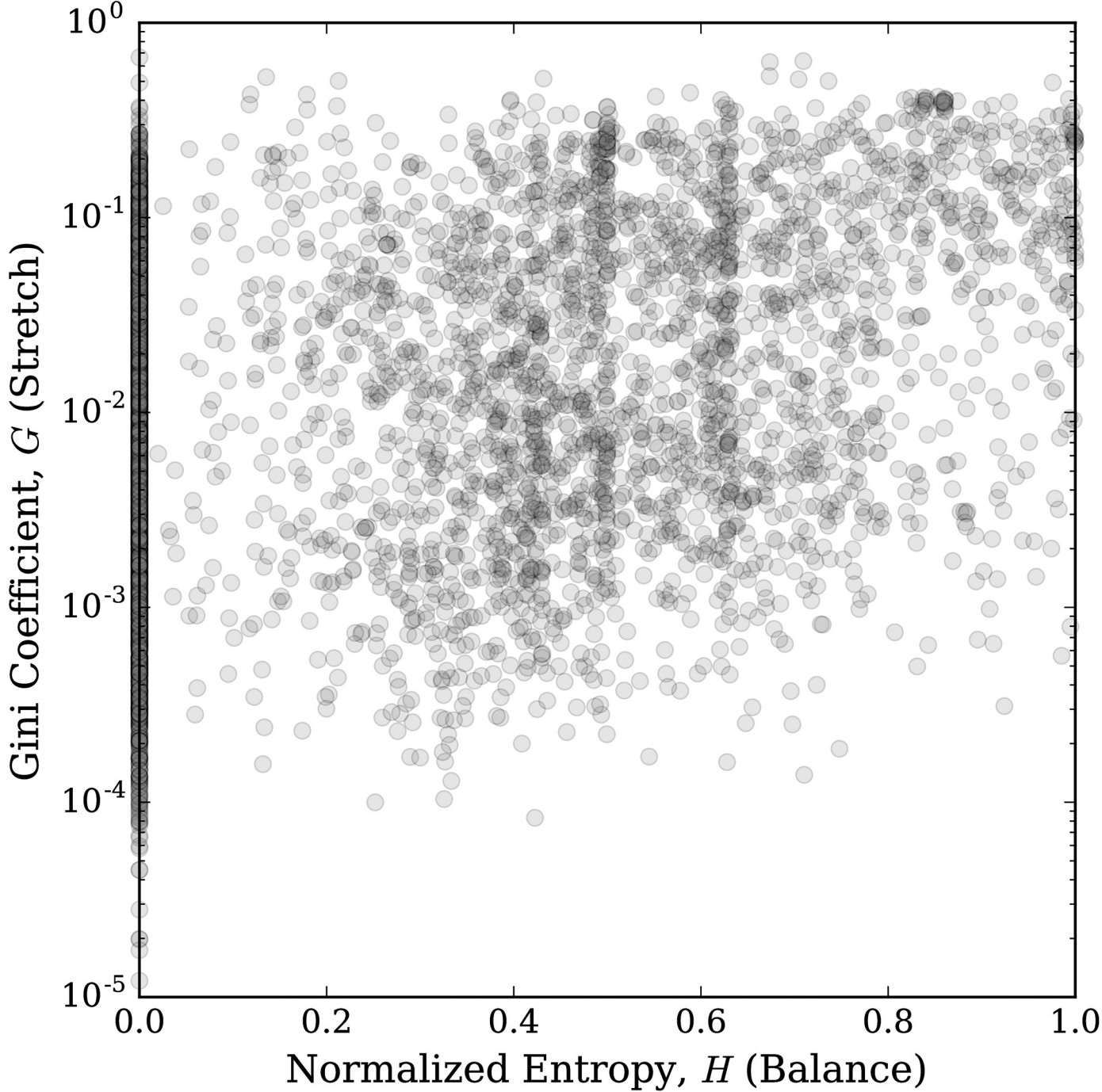

**Fig 9. Kernels plotted in balance-stretch parameter space.** Each kernel is plotted horizontally by the value of its balance parameter, given by normalized entropy, $H$, and vertically (on a logarithmic scale) by its stretch parameter, given by the Gini coefficient, $G$, of its token count distribution. Larger entropy implies greater balance and larger Gini coefficient implies greater stretch.

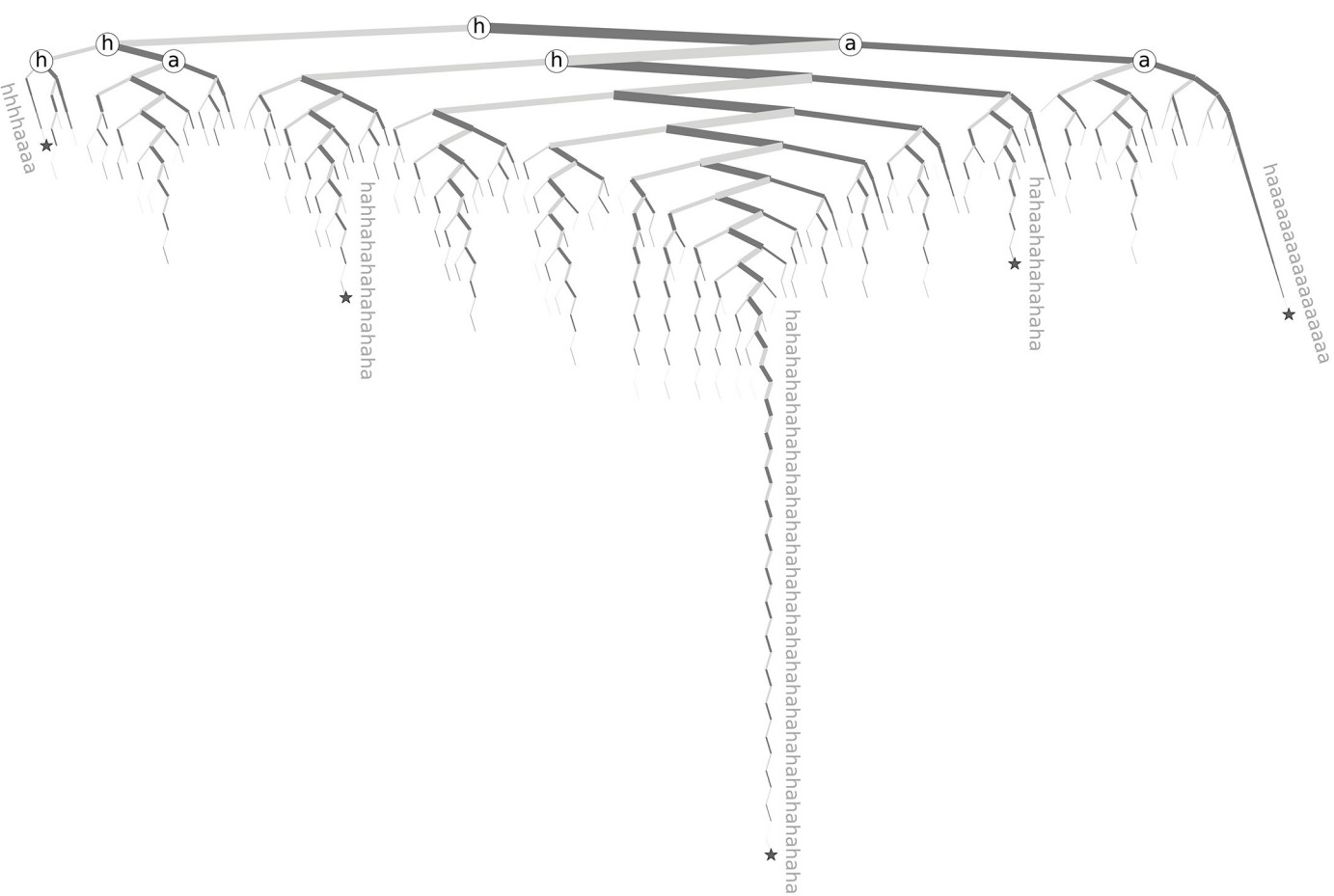

**Fig 10. Spelling tree for the kernel (ha).** The root node represents 'h'. From there, branching to the left (light gray edge) is equivalent to appending an 'h'. Branching to the right (dark gray edge) is equivalent to appending an 'a'. The edge width is logarithmically related to the number of tokens that pass along that edge when spelled out. A few example words are annotated, and their corresponding nodes are denoted with a star. This tree was trimmed by only including words with a token count of at least 10,000. The code used to create the figures for these spelling trees is largely based on the algorithm presented by Wetherel and Shannon [38]. We note that Mill has written a more recent paper based largely on this earlier work specialized for Python [39], and an implementation for it as well [40], but they both contain algorithmic bugs (detailed in S3 Appendix).

The spelling tree for (ha) has a number of interesting properties. Most notable among them is the self-similar, fractal-like structure. The main branch line dropping down just right of center represents the perfect alternating sequence 'hahahahaha. . .', as shown by the annotated example at the leaf of this line. There are also many similar looking subtrees that branch off from this main branch that each have their own similar looking main branch. These paths that follow the main branch, break off at one location, and then follow the main branch of a subtree represent words that are similar to the perfect alternating laugh, but either have one extra 'h' (if the branch veers left) or one extra 'a' (if the branch leads right). For example, the middle left annotation shows that the fourth letter was an extra 'h', and then the rest of the word retained an accurate alternating pattern. This word, 'hahhahahahahahaha', appeared 13,894 times in our dataset.

The tree also shows that 'haaaaa. . .' is a strong pattern, as can be seen farthest right in the (ha) spelling tree. The subtrees on the right show that users also start with the back and forth pattern for a stretch, and then finish the word with trailing 'a's. Many other patterns also

appear in this tree, and additional patterns are occluded by our trimming of the tree, but likely most of these come from users trying to follow one of the patterns we have already highlighted and introducing mistypings.

We made similar trees for every kernel that had a single occurrence of a two letter element, where the tree represents just the section of word that matches the two letter element. These trees are trimmed by only including words that have a token count of at least the fourth root of the total token count for the stretched tokens.

Fig 11 gives eight more examples of these spelling trees. The trees for (ja) and (xo) have many of the same characteristics as the tree for (ha), as do most of the trees for kernels that are a two letter element where tokens predominantly alternate letters back and forth. For the tree for (xo), the pattern where the first letter of the two letter element is stretched, followed by the second letter being stretched, such as 'xxxxxooooo', is more apparent, as seen by long stretches of just branching left followed by long stretches of just branching right. This type of pattern is even more notable in the trees for (aw), and especially (fu). The tree for (mo) has stretched versions for both 'mom' and 'moo'. Similarly, the tree for h(er) shows stretched versions of both 'her' and 'here', where we see that both 'e's and the 'r' all get stretched. In the tree for (to), the word 'totoo' has a much larger token count then words stretched beyond that (noticeable by the fact that the edges leaving that node are much smaller than the edge coming in). The word 'totoo' is Tagalog for 'true'. Finally, every example tree here does show the back and forth pattern to at least some extent. All of the trees created are available for viewing in Online Appendix D at http://compstorylab.org/stretchablewords/.

## 3.5 Mistypings

Mistypings appear often in tweets and we see evidence of them in stretched words. For example, the kernel n[o](io) is likely a result of mistypings of n[o]. On at least some platforms, holding down the key for a letter does not make that letter repeat, so one must repeatedly press the same key. For the standard QWERTY keyboard layout, the letter 'i' is next to the letter 'o', so it would be easy to accidentally press the letter 'i' occasionally instead of 'o' when trying to repeat it many times, especially on the small keyboards accompanying mobile phones. This sort of thing could lead to a kernel like n[o](io) when users try to stretch the word 'no'. Similarly, the letters 'a' and 's' are next to each other on a QWERTY keyboard, so a kernel like (ha)s(ha)(sh)(ah) likely comes from mistypings of the much simpler kernel (ha).

However, it is not always clearly apparent if a kernel is from mistypings or on purpose, or perhaps comes as a result of both. For example, the letter 'b' is close to the letter 'h', so the kernel (ha)b(ah) could come from mistypings of (ha). But, this form could also be intentional, and meant to represent a different kind of laughter. For example, (ba)(ha) is a highly ranked kernel (rank 211) representing a comedically sinister kind of laughter. Similarly, (ja) is a core component of laughter in Spanish, but 'j' is next to 'h' on the QWERTY keyboard, so it is not apparent if a kernel like (ha)j(ah)(ja)(ha) comes from mistypings or from switching back and forth between English and Spanish as the word stretches.

Our methodology may enable further study of mistypings. For example, Fig 12 shows the distribution, balance plot, and spelling tree for the kernel n[o](io). The distribution shows that it is not a strong kernel, with the lower rank of 4,858, compared to a rank of 8 for (no). The balance plot shows that the letter 'i' is not stretched much, and the spelling tree shows that the word is mostly just a repetition of 'o's. On the whole, the evidence suggests that the kernel n[o](io) is mainly a result of mistypings.

These tools can also be used to help study what are likely misspellings, rather than mistypings. For example, Fig 13 shows the spelling tree for the kernel hear(ta)ck (which does not

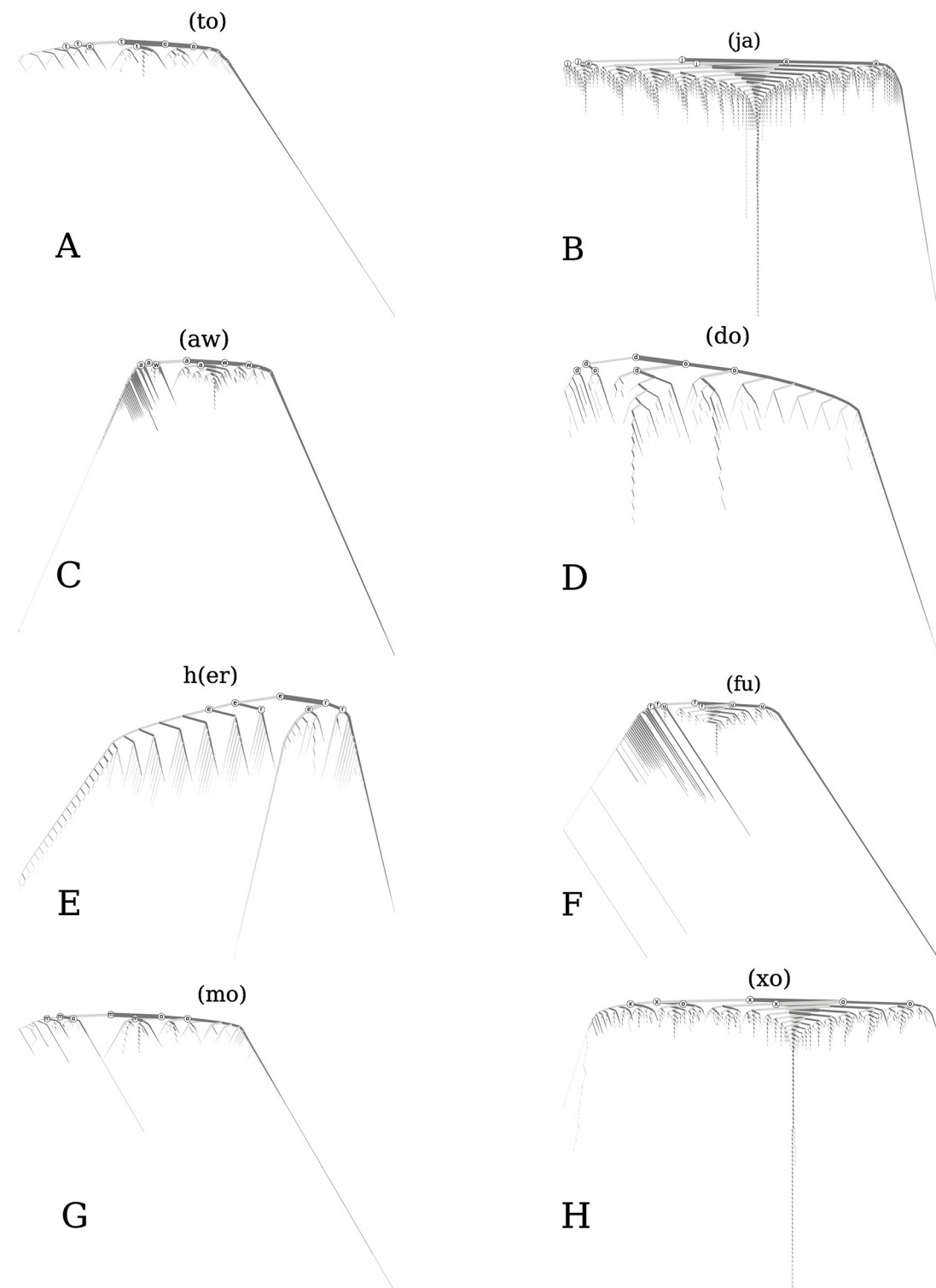

**Fig 11. A collection of example spelling trees.** From left to right, top to bottom, trees for the kernels (to), (ja), (aw), (do), h(er), (fu), (mo), and (xo).

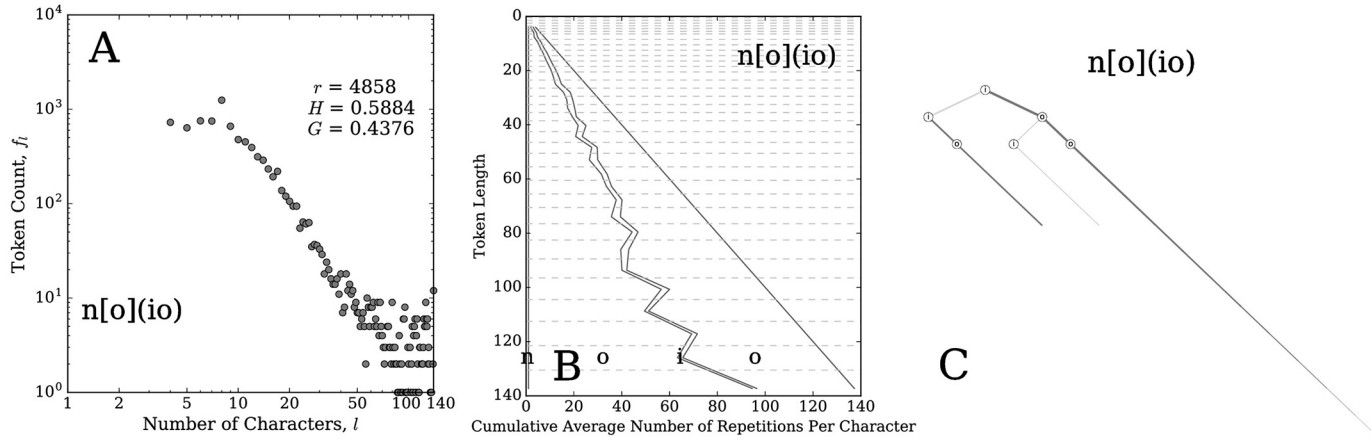

**Fig 12. (A) Token count distribution, (B) balance plot, and (C) spelling tree for the kernel n[o](io).** In general, these types of plots offer diagnostic help when studying mistypings. In this case, they provide evidence towards the conclusion that the words that match this kernel were likely meant to be stretched versions of the word 'no' with a few mistaken 'i's included. Note that 'i' is next to 'o' on a standard QWERTY keyboard.

actually fall within our rank cutoff, as described in Sec. 2, but provides a good example). The word 'attack' has two 't's. Thus, the word 'heartattack' (if written as one word; usually it is two) should, under normal spelling, have a double 't' after the second 'a'. From Fig 13 we can see from the weights of the branches that it is often written as 'heartatack', with a single 't' instead of the double 't'.

## 4 Concluding remarks

In this paper, we have studied stretched words, which are often used in spoken language. Until the advent of social media, stretched words were not prevalent in written language and largely absent from dictionaries. The area of stretchable language is rich, and we have discovered that these words span at least the two dimensional parameter space of balance and stretch.

As we mentioned in the beginning, there are many reasons why in spoken language people stretch a word, often done to increase the expressiveness of the word. When stretched words first started showing up in written forms of communication, they seemed to be mainly a direct written representation of spoken stretched words [6, 8] and even the few that showed up in literature mainly showed up during the dialogue in fiction, again showing this direct correspondence to spoken language [8]. Over time, these forms of expressive written language have become more common, especially in less formal contexts and with younger users [5]. Even this reflects spoken language, where we have developed gestures and expressions and other nonverbal visual clues that we use during our informal speech that are not seen as much in more formal speech contexts, as used by, for example, traditional news anchors [8].

As they have become more common, there is evidence that written stretchable words have begun to take on a life of their own, and are losing some of their direct connection to their spoken counterpart [6, 8, 26]. One clue to this is which letters are stretched. In their main study on email messages, Kalman and Gergle found that most of the stretching was articulable, but in a small exploratory study at the end with blog posts they found an increase in stretching that is inarticulable and also especially found a general increase in the stretching of the last letter of words [6].

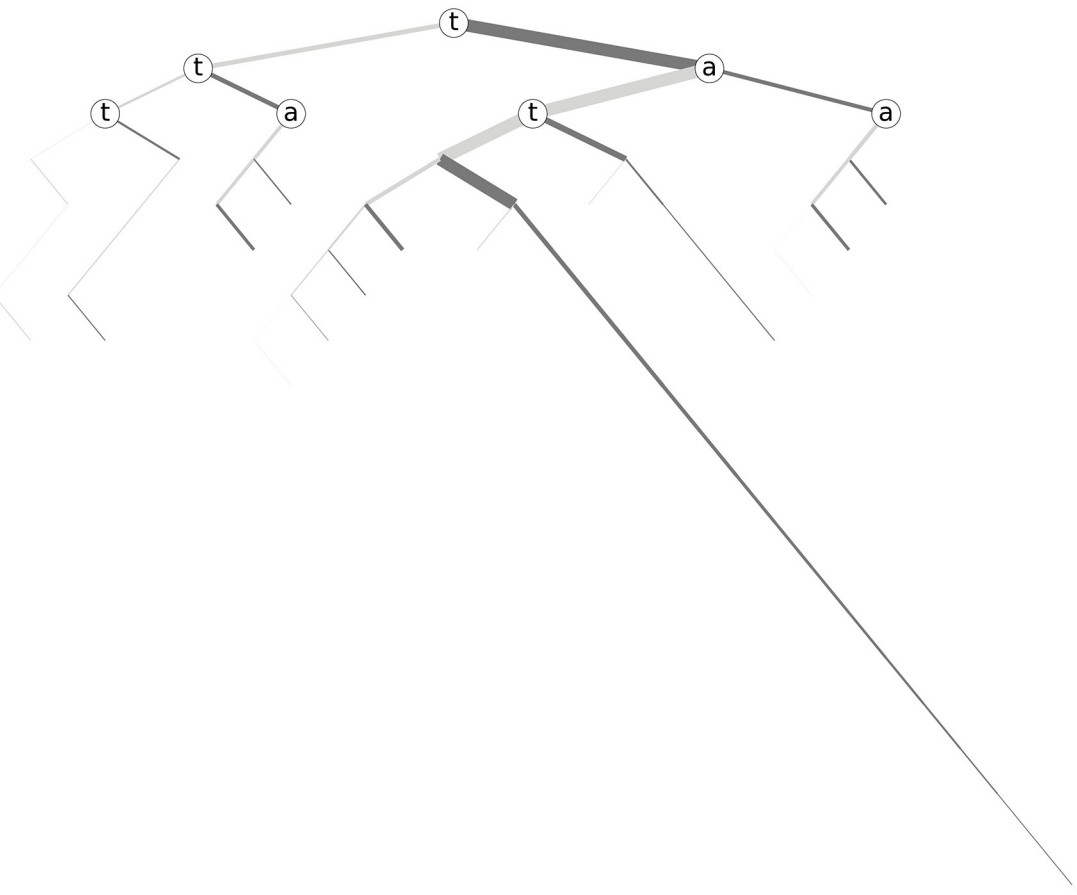

**Fig 13. Spelling tree for the kernel hear(ta)ck.** From this tree, we can see the relative number of times the word 'heartatack' is written rather than 'heartattack', indicating a common misspelling.

An initial look at our findings supports these studies. Our data covers from the earlier days of Twitter through 2016 and thus likely includes any changes in the use of stretchable words. We certainly see, especially from the balance plots, that articulable parts of words get stretched a lot, as if mimicking spoken stretching, but we also see stretching that is inarticulable. For example, looking at Fig 5, we see that the plosive 'g' is not stretched as much as the articulable ones, but it certainly exists. Furthermore, the balance plots for [p][l][e][a][s][e] and [h][e][l][p] (available in Online Appendix C) show a lot of stretching for the final 'e' and 'p' even though both are inarticulable. In particular, for 'please' the final 'e' is by far the most stretched letter. This aligns with what Kalman and Gergle found when looking at stretching 'please' and 'help' in blogs [6].

This evidence, along with the initial findings of others, suggests that stretchable words have grown to be more of a visual cue as well. Stretched words tend to stand out more. Some tweets are comprised of a single stretched word. We also saw tweets where the author was pleading to a particular celebrity, asking that celebrity to follow them, where every letter of the tweet was

stretched, in an apparent attempt to stand out and be noticed. We did see from the jellyfish plots of balance in Sec. 3.2 that the highest ranked kernels, that is, the kernels that are stretched the most often, tended to be more balanced than average.

It would be interesting to study these things further, such as the distinction between visual and phonetic stretching or how the patterns of stretching have changed over time or differ across geographic regions. We have developed the methods here that would allow for a much more in depth study of these and other linguistic research questions. Looking at what parts of words, such as the end of words, or which letters, or class of letters (e.g., comparing vowels and consonants or stops and fricatives) get stretched more, would also be interesting. Other studies could include comparing stretching across different grammatical or semantic classes of words or looking at changes in stretching patterns across different communication media (e.g., comparing Twitter and emails).

The tools we have developed not only help uncover the hidden dynamics of stretchable words, but can be further applied to study phenomena such as mistypings and misspellings, and possibly more. Online dictionaries, such as the Wiktionary [41], could use our kernels as a general entry for each type of stretchable word, and include the balance and stretch parameters as part of their structured word information, as they do, for example, with part of speech.

Searching for stretched words, as we discovered during the course of our research for this study, is not easy. Search engines do not do well with stretched words. They may be able to find a specific word that is given to them, but if trying to find stretched versions of a particular word in general, they suffer. Again, the use of our kernels could help here as a more general way of searching and indexing.

It is known that natural language processing (NLP) can be hard with social media because of the nonstandard language that is often used [4, 21–23]. Natural language processing software and toolkits could use the techniques we developed to help with processing stretched words. For example, stretched words could first be distilled to their kernels, and the base word could be extracted from that. Then other processing, such as part of speech tagging, could be applied to the base word. Similarly, spell checking software may be able to use our methods to help prevent marking stretched words as misspellings. Our procedures could also be used to help prevent typosquatting [42]. Twitter could use our methods to help improve their spam filter, looking for slight variations of tweets. Also, spelling trees could more generally be used to analyze the construction of any sequence, such as genome sequences.

However, much more could be done. We have restricted our study to words containing only Latin letters. Future work could extend this to include all characters, including punctuation and emojis. We also limited the way we constructed kernels, focusing only on one and two letter elements. This can be expanded to three letter elements and possibly beyond to capture the characteristics of words like 'omnomnomnom'. Furthermore, our methodology for creating kernels leads to situations where, for example, we have both (ha)g(ah) and (ha)(ga)(ha) as kernels. Expanding to three letter elements and beyond in the future could collapse these forms, and related kernels, into a kernel like (hag).

Along with more advanced kernels, similar but more advanced spelling trees could be developed. We only created spelling trees for kernels with a single two letter element. Future work could explore kernels with more than two letter elements. They could also be created for every kernel, where the branching of even the single letter elements is shown, where one branch would signify the repetition of that letter and the other branch would signify moving onto the next letter of the kernel. Furthermore, to go with three letter elements, ternary trees could be developed. Among other things, this would reveal mistypings like (ha)(hs), for example, if this became a kernel with a three letter element like (has), and we assume that the 's' is

mostly a mistyping of the letter 'a' in the kernel (ha). This situation should be discernible from the case where the word 'has' is stretched.

Finally, our methodology could be used to explore linguistic and behavioral responses to changes in Twitter's protocol (e.g., character length restrictions) and platform (e.g., mobile vs. laptop). For example, what are the effects of auto-correct, auto-complete, and spell check technologies? And what linguistic changes result from platform restrictions such as when a single key cannot be held down anymore to repeat a character? Also, we only considered tweets before the shift from the 140 to 280 character limit on Twitter. Some initial work indicates that the doubling of tweet length has removed the edge effect that the character limit creates [43]. Further work could study how this change has affected stretchable words, and in particular, the tail of their distributions.

## Supporting information

**S1 Appendix. Alternate balance measure.**
(PDF)

**S2 Appendix. Stretch ratio.**
(PDF)

**S3 Appendix. "Drawing presentable trees" algorithmic bugs.**
(PDF)

## Acknowledgments

The authors would like to posthumously thank Margaret Lima for all her help and support of TJG during the early stages of this research and general collaboration between the present authors.

## Author Contributions

**Conceptualization:** Tyler J. Gray, Christopher M. Danforth, Peter Sheridan Dodds.

**Data curation:** Tyler J. Gray.

**Formal analysis:** Tyler J. Gray, Christopher M. Danforth, Peter Sheridan Dodds.

**Investigation:** Tyler J. Gray.

**Methodology:** Tyler J. Gray, Christopher M. Danforth, Peter Sheridan Dodds.

**Software:** Tyler J. Gray.

**Supervision:** Christopher M. Danforth, Peter Sheridan Dodds.

**Visualization:** Tyler J. Gray.

**Writing – original draft:** Tyler J. Gray.

**Writing – review & editing:** Tyler J. Gray, Christopher M. Danforth, Peter Sheridan Dodds.

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
