## [Decision Letter · Decision Letter 0]

3 Sep 2019

PONE-D-19-19332

Hahahahaha, Duuuuude, Yeeessss!: A two-parameter characterization of stretchable words and the dynamics of mistypings and misspellings

PLOS ONE

Dear Mr. Gray,

Thank you for submitting your manuscript to PLOS ONE. After careful consideration, we feel that it has merit but does not fully meet PLOS ONE’s publication criteria as it currently stands. Therefore, we invite you to submit a revised version of the manuscript that addresses the points raised during the review process. 

We would appreciate receiving your revised manuscript by Oct 18 2019 11:59PM. To enhance the reproducibility of your results, we recommend that if applicable you deposit your laboratory protocols in protocols.io, where a protocol can be assigned its own identifier (DOI) such that it can be cited independently in the future. For instructions see: http://journals.plos.org/plosone/s/submission-guidelines#loc-laboratory-protocols

We look forward to receiving your revised manuscript.

Kind regards,

Lidia Adriana Braunstein, Phd in Physics

Academic Editor

PLOS ONE

Journal Requirements:

2. In your Methods section, please include additional information about your dataset and ensure that you have included a statement specifying whether the collection method complied with the terms and conditions for the website.

Reviewers' comments:

Reviewer's Responses to Questions

**Comments to the Author**

1. Is the manuscript technically sound, and do the data support the conclusions?

Reviewer #1: Yes

Reviewer #2: Yes

2. Has the statistical analysis been performed appropriately and rigorously? 

Reviewer #1: Yes

Reviewer #2: Yes

3. Have the authors made all data underlying the findings in their manuscript fully available?

Reviewer #1: Yes

Reviewer #2: Yes

4. Is the manuscript presented in an intelligible fashion and written in standard English?

Reviewer #1: Yes

Reviewer #2: Yes

5. Review Comments to the Author

Reviewer #1: This is a very charming piece of work; quite apart from its appeal, it is also a very nice contribution to the study of language use online. It is the first time I've seen this phenomenon studied in a rigorous fashion. There is also some really nice technical innovation in the appendix.

I have only one minor point: we know that regressions on a log-log plot are not statistically robust measures of power law indices. Can the authors try, e.g., the Clauset-Shalizi powerlaw estimation tools? These allow for an xmin specification, and will not be sensitive to extreme outliers on the right hand side, so should work well.

Congratulations on a very nice paper.

Reviewer #2: Overall, I really enjoyed the paper and I think it should be accepted for publication after some minor revisions, although the authors might want to go a bit further then what I’m insisting on, which isn’t much. The methods and findings are interesting, original, and clearly presented. It was a fun paper to read and I feel like it provides a real foundation for further linguistic analysis.

As a linguist, I do find the framing, analysis, and discussion all a bit superficial though. Right now this reads solely as a methodological paper, but I think there is more insight here than what was discussed. The rationale for the paper is almost entirely around the application of these methods, and even there it’s pretty limited, mostly lexicography and general NLP, and the NLP examples aren’t really clear to me. This is fine – the author’s aren’t overclaiming, which is nice – but there is some more room here for discussion if they want.

The introduction/literature review is especially quick. I realise the authors aren’t linguists, but my one insistence is that the authors review some previous research, if only briefly. The authors say lengthening is fundamental to speech, and I guess that’s true, but that’s really all they say. That point could be developed a bit, but at the very least the processes of vowel lengthening and gemination should be noted, and the directly relevant literature in linguistics and NLP should be acknowledged. Like in a few minutes online I found a bunch of very relevant research, which should be cited at a minimum:

https://repository.upenn.edu/pwpl/vol18/iss2/14/

https://www.aclweb.org/anthology/D11-1052

https://www.sciencedirect.com/science/article/pii/S0747563214000594

https://gretchenmcculloch.com/book/

https://www.aclweb.org/anthology/N13-1037

There is other material the authors might find relevant as well, if they want to go further. The frequency distribution of word lengths obsewrved by Zipf, for example, is not wholly tangential to the topic of this paper. We have also done research on new words on Twitter in particular that has looked at creative spellings, including stretched words, and at how much word length matters in terms of predicting the success of new words over time.

https://doi.org/10.1017/S1360674316000113

http://evolang.org/torun/proceedings/paperpdfs/Evolang_12_paper_171.pdf

In their discussion/conclusion, I think the authors should also consider how their methods and results could inform our understanding of how and why words are stretched on Twitter. To me that is the main scientific value of this method, and it isn’t really realised. A number of relevant observations are made as the methods are described, but they are never really brought together. And then of course they aren’t related back to previous research, since none is covered. I’m not insisting on this. I’m fine with a purely methodological paper. It just seems to me like a missed opportunity. I think that would increase the impact and significance of the paper.

Otherwise, in terms of the methods, I think they are well presented for the most part. I don’t find the prose descriptions especially easy to follow. On the one hand, I think they could be explained a bit more plainly and in some more detail. On the other hand, equations/algorithm descriptions in the main text would help clarify things. After I look at the figures and read the captions, I understand the methods, but I was a bit lost up till then, but maybe that’s just me. FWIW the jellyfish plots were especially unclear to me. I more or less get the idea, but I’m not sure I’m 100% totally following there. And anyway I don’t really see much point to them – like why are they useful? I’d recommend the authors either expand or cut that section, but it’s a pretty minor point and I’m not bothered one way or the other.

Also, briefly in terms of the plots, the authors might think about including character labels on the lines in the balance plots, so that they can be read without a caption. I’m also not sure how much varying the shading and the width of the branches in the spelling trees. It’s a neat effect, but I found it kind of hard to read, especially as the trees got more complex and the resolution got smaller. Plus, if I understand correctly, it’s all redundant information. Anyway, just my reaction. I realise some thought has been put into this, and it’s not a big deal.

Overall, I find the paper really interesting and it seems like the methods could be used for lots of different types of analyses. I can think about a bunch of interesting linguistic research questions that one could pursue with the methods: looking for differences in stretching across words classified according to their grammatical or semantic characteristics; looking for change over time or across space the spelling of specific words; looking at lengthening patterns on different character types (e.g. vowels vs. consonants; stops vs. fricatives).

6. PLOS authors have the option to publish the peer review history of their article (what does this mean?). If published, this will include your full peer review and any attached files.

Reviewer #1: No

Reviewer #2: Yes: Jack Grieve

---

## [Author Response · Author response to Decision Letter 0]

13 Mar 2020

Dear Reviewers,

We thank the reviewers for their time and effort spent reviewing our manuscript and appreciate their comments and suggestions towards improving it. We also thank both reviewers for their kind general remarks about our manuscript and are glad you seemed to enjoy the paper. As detailed further below, we have worked to address your comments and hope we have a clearer, overall improved article as a result.

Reviewer 1

Comment:

“We know that regressions on a log-log plot are not statistically robust measures of power law indices. Can the authors try, e.g., the Clauset-Shalizi powerlaw estimation tools? These allow for an xmin specification, and will not be sensitive to extreme outliers on the right hand side, so should work well.”

Response:

We agree that those methods are good for finding power law indices. However, we do not find any power laws in our work. We mention that the token count distributions seem to follow a rough power law shape, but we do not calculate any indices or formally test how well a power law fits. We do perform linear regression on a log-log plot in relation to Fig. 2 when we are calculating a cutoff rank. However, we do not mean to imply that the underlying distribution follows a power law. We just use the line as a step in our attempt to find a cutoff in a more principled way than arbitrarily setting a cutoff using the idea of a cutoff frequency from signals analysis. In this way, the line should almost be thought of more like a roughly tangent line to the main part of the curve, giving us something to calculate a drop from. Furthermore, the exact cutoff is not important and will not change our results in any meaningful way. We have added the following text to the paper in an attempt to make this clearer to the reader:

“Note that we are using a rough guide to find a practical cutoff for the number of kernels we include in our study. While we are finding a linear fit as part of this process, this token count distribution is not some archetypal power-law. We merely use the regression line as a reference from which to calculate a drop analogous to the process of finding a cutoff frequency, and the precise cutoff is not particularly important. The cutoff rank is not used in the statistics of any individual kernel, and for the analyses that examine how stretchable words behave as a function of kernel rank, the resultant figures and statistics will only be affected at the margin of the cutoff rank. An alternative might be to simply pick a cutoff rank based on visual inspection of Fig. 2 or to pick a lower bound for the data amount (token count sum), and find which rank falls below that bound.”

Reviewer 2

Comment:

“The introduction/literature review is especially quick. I realise the authors aren’t linguists, but my one insistence is that the authors review some previous research, if only briefly. The authors say lengthening is fundamental to speech, and I guess that’s true, but that’s really all they say. That point could be developed a bit, but at the very least the processes of vowel lengthening and gemination should be noted, and the directly relevant literature in linguistics and NLP should be acknowledged. Like in a few minutes online I found a bunch of very relevant research, which should be cited at a minimum:

https://repository.upenn.edu/pwpl/vol18/iss2/14/

https://www.aclweb.org/anthology/D11-1052

https://www.sciencedirect.com/science/article/pii/S0747563214000594

https://gretchenmcculloch.com/book/

https://www.aclweb.org/anthology/N13-1037”

Response:

We added a brief mention of vowel lengthening and gemination, but do not belabor it as we feel it is quite separate from the lengthening we are examining in our study. We appreciate the links to the related research and have reviewed and referenced all of them in our revision, and also included some more that we found.

Comment:

“As a linguist, I do find the framing, analysis, and discussion all a bit superficial though. Right now this reads solely as a methodological paper, but I think there is more insight here than what was discussed. The rationale for the paper is almost entirely around the application of these methods, and even there it’s pretty limited, mostly lexicography and general NLP, and the NLP examples aren’t really clear to me. This is fine – the author’s aren’t overclaiming, which is nice – but there is some more room here for discussion if they want.

There is other material the authors might find relevant as well, if they want to go further. The frequency distribution of word lengths observed by Zipf, for example, is not wholly tangential to the topic of this paper. We have also done research on new words on Twitter in particular that has looked at creative spellings, including stretched words, and at how much word length matters in terms of predicting the success of new words over time.

https://doi.org/10.1017/S1360674316000113

http://evolang.org/torun/proceedings/paperpdfs/Evolang_12_paper_171.pdf

In their discussion/conclusion, I think the authors should also consider how their methods and results could inform our understanding of how and why words are stretched on Twitter. To me that is the main scientific value of this method, and it isn’t really realised. A number of relevant observations are made as the methods are described, but they are never really brought together. And then of course they aren’t related back to previous research, since none is covered. I’m not insisting on this. I’m fine with a purely methodological paper. It just seems to me like a missed opportunity. I think that would increase the impact and significance of the paper.”

Response:

Thank you for the suggestions on how you think we could make our paper more impactful. In the revised manuscript, we try to include a clearer application of our methods to NLP and we include references to the provided papers. In the section of the paper where we discuss the distributions we added a paragraph discussing Zipf’s brevity law and how it relates. We have also significantly increased the discussion in the concluding remarks section of the paper, bringing together some results from our paper and some from the research of others. Though, perhaps, we have not done so as much as you think there is opportunity for. We leave the remainder of this to further research.

Comment:

“Otherwise, in terms of the methods, I think they are well presented for the most part. I don’t find the prose descriptions especially easy to follow. On the one hand, I think they could be explained a bit more plainly and in some more detail. On the other hand, equations/algorithm descriptions in the main text would help clarify things. After I look at the figures and read the captions, I understand the methods, but I was a bit lost up till then, but maybe that’s just me. FWIW the jellyfish plots were especially unclear to me. I more or less get the idea, but I’m not sure I’m 100% totally following there. And anyway I don’t really see much point to them – like why are they useful? I’d recommend the authors either expand or cut that section, but it’s a pretty minor point and I’m not bothered one way or the other.”

Response:

We tried to find the parts of the paper that were potentially most confusing and tried to make them clearer, including adding equations in quite a few places. In particular, we also added more explanation around the jellyfish plots explaining their usefulness.

Comment:

“Also, briefly in terms of the plots, the authors might think about including character labels on the lines in the balance plots, so that they can be read without a caption. I’m also not sure how much varying the shading and the width of the branches in the spelling trees. It’s a neat effect, but I found it kind of hard to read, especially as the trees got more complex and the resolution got smaller. Plus, if I understand correctly, it’s all redundant information. Anyway, just my reaction. I realise some thought has been put into this, and it’s not a big deal.”

Response:

For the balance plots, we had considered adding the character labels to the plots, but hadn’t largely because for many of the plots there is not room and the labels would overlap. However, for the plots in the main paper, that is not really the case. What we decided to do is only add character labels for characters that are allowed to stretch in the kernel, and then only if they do not overlap each other when printed on the plot.

For the spelling trees, the width of the branches are not redundant information. The width is related to the number of tokens that pass through that branch when spelled out. So wider branches reflect paths that are more common when spelling out stretched words, and this information is not available in any other way from the figure. The shading however is indeed redundant information, just reflecting the direction of the branch. In an earlier life, the trees were all a single color. Through our iterations of making the figures though, we felt that the two shades helped with following some of the patterns and have decided to keep the shading. Even though we did not change them this time, it is still always useful to get feedback on the figures and we much appreciate it.

Comment:

“Overall, I find the paper really interesting and it seems like the methods could be used for lots of different types of analyses. I can think about a bunch of interesting linguistic research questions that one could pursue with the methods: looking for differences in stretching across words classified according to their grammatical or semantic characteristics; looking for change over time or across space the spelling of specific words; looking at lengthening patterns on different character types (e.g. vowels vs. consonants; stops vs. fricatives).”

Response:

Thank you for the additional future research suggestions. We have added these to the future research part of our concluding remarks.

Yours sincerely and on behalf of the manuscript’s authors,

Tyler Gray

Department of Mathematics and Statistics 

The University of Vermont

---

## [Decision Letter · Decision Letter 1]

27 Apr 2020

Hahahahaha, Duuuuude, Yeeessss!: A two-parameter characterization of stretchable words and the dynamics of mistypings and misspellings

PONE-D-19-19332R1

Dear Dr. Gray,

We are pleased to inform you that your manuscript has been judged scientifically suitable for publication and will be formally accepted for publication once it complies with all outstanding technical requirements.

With kind regards,

Lidia Adriana Braunstein, Phd in Physics

Academic Editor

PLOS ONE

Additional Editor Comments (optional):

Reviewers' comments:

Reviewer's Responses to Questions

**Comments to the Author**

1. If the authors have adequately addressed your comments raised in a previous round of review and you feel that this manuscript is now acceptable for publication, you may indicate that here to bypass the “Comments to the Author” section, enter your conflict of interest statement in the “Confidential to Editor” section, and submit your "Accept" recommendation.

Reviewer #2: All comments have been addressed

2. Is the manuscript technically sound, and do the data support the conclusions?

Reviewer #2: Yes

3. Has the statistical analysis been performed appropriately and rigorously? 

Reviewer #2: Yes

4. Have the authors made all data underlying the findings in their manuscript fully available?

Reviewer #2: Yes

5. Is the manuscript presented in an intelligible fashion and written in standard English?

Reviewer #2: Yes

6. Review Comments to the Author

Reviewer #2: Paper looks really good! I always really liked the study and I think the background and methods are much clearer now. Sorry about being a bit slow with the review. Busy these days.

7. PLOS authors have the option to publish the peer review history of their article (what does this mean?). If published, this will include your full peer review and any attached files.

Reviewer #2: Yes: Jack Grieve

---

## [Editor Report · Acceptance letter]

4 May 2020

PONE-D-19-19332R1 

Hahahahaha, Duuuuude, Yeeessss!: A two-parameter characterization of stretchable words and the dynamics of mistypings and misspellings 

Dear Dr. Gray:

I am pleased to inform you that your manuscript has been deemed suitable for publication in PLOS ONE. Congratulations! Your manuscript is now with our production department. 

With kind regards,

on behalf of

Dr. Lidia Adriana Braunstein 

Academic Editor

PLOS ONE